# Towards Sub-Second Molecular Docking as a Structural Primitive: A Quantized Consistency Diffusion Framework

**Kexin Zhang** [1 2 3] **Weichen Qin** [1] **Yue Teng** [1] **Jiale Yu** [1] **Yuanyuan Ma** [4 3] **Jinyu Lin** [4 3] **Liping Sun** [5] **Jie Zheng** [1] **Jingyi Yu** [1]

## Abstract

Agent-centered scientific discovery is turning scientific models into always-on computational infrastructure. In this paradigm, AI agents coordinate tools, interpret feedback, and drive high-frequency research loops, requiring domain models that are both accurate and callable in real time. Molecular docking exposes this bottleneck: it provides essential structural feedback for drug discovery, yet current high-fidelity docking and co-folding models remain limited by iterative generative refinement and heavy computation. We present a compute-efficient co-folding framework that turns molecular docking into a sub-second structural primitive. Because docking methods operate under different levels of structural prior, we report accuracy under information-level-matched protocols, comparing blind settings with blind generative methods and interface-informed settings with surface- or interface-informed baselines. Our framework combines two ideas. First, Progressive Consistency Regularization (PCR) compresses diffusion dynamics into reliable few-step inference through reconstruction-anchored consistency tuning. Second, Residual-Safe Quantization preserves high-fidelity residual streams and geometry-sensitive operations in BF16 while quantizing selected compute-intensive linear transformations. Our model achieves state-of-the-art docking accuracy under the matched interface-informed protocol, reports blind docking performance separately under the matched blind protocol, and generates five conformations for a representative 256-token complex in 0.17 seconds on a single NVIDIA H20 GPU, delivering a $> 300\times$ speedup over AlphaFold3 under the benchmarked setting. Together, these results move molecular docking from an offline generative simulator toward a real-time structural primitive for agent-centered drug discovery.

## 1. Introduction

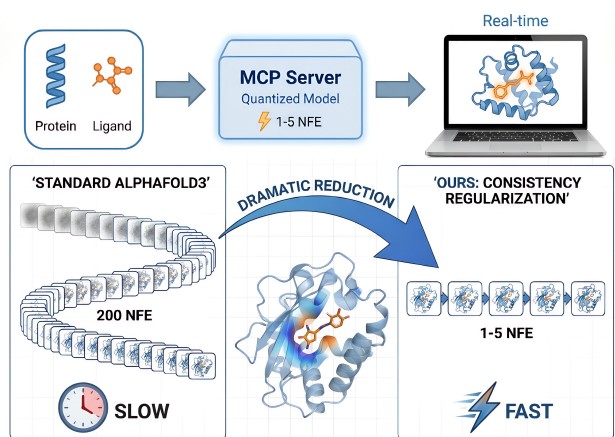

*Figure 1.* **Bridging the Gap to Real-Time Docking.** By combining PCR for few-step sampling with Residual-Safe Quantization for deployment efficiency, our model achieves sub-second inference on a single NVIDIA H20 GPU. Exposed through an Model Context Protocol (MCP) server or similar tool interface, it can serve as a high-throughput, low-latency structural primitive inside agent-centered discovery workflows.

Molecular docking is a central task in structure-based drug discovery (SBDD) (Anderson, 2003; Schneuing et al., 2024). Given a protein pocket and a candidate ligand, it predicts where the ligand binds and what three-dimensional interface it forms. This structural feedback guides virtual screening, lead optimization, and protein–ligand design (Rester, 2008;

---

[1]School of Information Science and Technology, ShanghaiTech University, Shanghai, China [2]Lingang Laboratory, Shanghai, China [3]ProteinDance Co., Ltd., Shanghai, China [4]School of Life Science and Technology, ShanghaiTech University, Shanghai, China [5]iHuman Institute, ShanghaiTech University, Shanghai, China. Correspondence to: Liping Sun <sunlp@shanghaitech.edu.cn>, Jie Zheng <zhengjie@shanghaitech.edu.cn>, Jingyi Yu <yujingyi@shanghaitech.edu.cn>.

*Proceedings of the $43^{rd}$ International Conference on Machine Learning*, Seoul, South Korea. PMLR 306, 2026. Copyright 2026 by the author(s).

Maia et al., 2020; Tinberg et al., 2013), making docking a core feedback mechanism for molecular discovery.

The field has made major progress in accuracy. Classical search-and-scoring pipelines rely on simplified scoring functions (Friesner et al., 2004; Halgren et al., 2004; Trott & Olson, 2010), restricted conformational search, and limited treatment of protein flexibility. Recent deep learning methods have pushed docking from approximate pose search toward high-fidelity structure generation (Corso et al., 2022; Cao et al., 2025). At the frontier of this shift are co-folding models, which jointly model protein–ligand complexes and better capture global binding topology, protein–ligand contacts, and local atomic geometry (Abramson et al., 2024a; Chai Discovery et al., 2024; Wohlwend et al., 2024; Passaro et al., 2025; Protenix Team et al., 2026; Zhou et al., 2025). Yet accuracy alone does not make a deployable docking engine. High-fidelity co-folding models rely on large pair representations, deep attention blocks, and iterative generative refinement. These mechanisms improve structural quality, but they also make inference sequential, costly, and difficult to scale. This creates a sharp accuracy–latency gap: co-folding models are making docking more accurate, but not yet efficient enough to become screening-scale infrastructure.

Evaluation adds a second complication. Modern docking systems do not operate under a single information regime. Some methods perform blind docking with minimal pocket information, whereas others use specified pockets, protein surfaces, residue-level constraints, or interface-informed priors. Comparing these settings without distinction can conflate model quality with input prior strength. We therefore treat input information as part of the evaluation protocol. Blind docking settings are compared with blind generative methods, while strong-prior settings are compared with surface- or interface-informed baselines. This information-level-matched view makes the accuracy comparison explicit and separates docking capability from the amount of structural context provided at inference time.

The need for fast docking is amplified by agent-centered scientific discovery. Large language models and scientific agents are beginning to coordinate tools, generate molecular hypotheses, interpret feedback, and iterate with human researchers (Wang et al., 2023; Lu et al., 2026). In such loops, docking is no longer an offline module at the end of a pipeline. It becomes a repeatedly invoked structural skill for ranking candidates, revising hypotheses, and exploring chemical space. In this regime, latency is not a minor engineering detail; it determines whether docking can remain inside an active reasoning loop.

In this work, we address this accuracy–latency gap. Our goal is not to build a heavier co-folding model, but to turn high-fidelity molecular docking into a sub-second structural primitive. The guiding principle is simple: preserve the computation that determines structural fidelity, and shorten the computation that limits deployment.

We first revisit the generative design of co-folding models. Since iterative sampling is the dominant algorithmic bottleneck, we ask which generative trajectory is most amenable to low number-of-function-evaluations (NFE) sampling. Under controlled comparisons with matched training and sampling factors, we find that few-step molecular generation is not only a matter of fewer solver steps, but also a matter of trajectory geometry. The Linear parameterization (Liu et al., 2022; Esser et al., 2024) provides a more discretization-friendly transport target and update rule than cosine/EDM-style denoising parameterizations (Nichol & Dhariwal, 2021; Karras et al., 2022). This motivates a streamlined design: instead of adding more modules, we compress the generative dynamics and the inference path.

Our framework contains two key components. First, we introduce **Progressive Consistency Regularization (PCR)**, a reconstruction-anchored consistency strategy (Song et al., 2023) that turns diffusion sampling into reliable few-step inference while preserving structural supervision. Second, we introduce **Residual-Safe Quantization**, a mixed-precision deployment strategy that keeps high-fidelity residual streams and geometry-sensitive operations in BF16 while quantizing selected compute-intensive linear transformations inside Pairformer and token-level Diffusion Transformer (DiT) blocks (Peebles & Xie, 2022). Together, these components transform molecular docking from a slow generative simulator into a real-time structural skill.

Our model achieves state-of-the-art docking accuracy under the matched interface-informed protocol, remains competitive under blind docking evaluation, and reaches sub-second inference under the same input-shape deployment protocol. It establishes a new accuracy–latency frontier for high-throughput virtual screening and agent-centered drug discovery.

Our contributions are summarized as follows:

- We identify **screening-scale callability** as the missing requirement for high-fidelity molecular docking: co-folding models are accurate, but not yet fast enough to serve as deployable infrastructure.

- We formalize **information-level-matched evaluation** for docking, separating blind docking settings from strong-prior surface- or interface-informed settings so that accuracy comparisons reflect comparable input information.

- We show that **few-step molecular generation is a trajectory-geometry problem**, and that Linear parameterization provides a more discretization-friendly tar-

get and update rule than cosine/EDM-style denoising parameterizations under controlled comparisons.

- We propose **Progressive Consistency Regularization (PCR)** to turn diffusion-based docking into reliable few-step inference without removing reconstruction supervision.

- We propose **Residual-Safe Quantization** to preserve geometry-sensitive residual streams while accelerating compute-heavy transformations, enabling sub-second docking with strong pose accuracy.

## 2. Related Work

### 2.1. Molecular Docking: From Approximate Search to High-Fidelity Co-folding

Molecular docking has long been a core tool in SBDD. Classical docking pipelines formulate the task as conformational search followed by empirical or physics-inspired scoring (Trott & Olson, 2010; Halgren et al., 2004; Friesner et al., 2004; Verdonk et al., 2003). These methods enabled practical virtual screening, but their accuracy is limited by simplified scoring functions, restricted ligand and protein flexibility, and approximate treatment of induced fit.

Deep learning has reshaped this landscape. Early models treated docking as direct geometric prediction. EquiBind (Stärk et al., 2022) and TANKBind (Lu et al., 2022) predicted ligand poses or protein–ligand contacts efficiently, but often struggled with local physical validity and fine-grained atomic geometry. Later methods improved molecular representation and interaction modeling through pre-training, attention, or interaction-aware architectures (Alcaide et al., 2024; Lai et al., 2024). A second line of work introduced generative docking. DiffDock (Corso et al., 2022) modeled ligand pose prediction as diffusion over rigid-body transformations and torsions, while Surf-Dock (Cao et al., 2025) incorporated surface-level geometry to improve interface recognition.

The recent frontier is co-folding. Models such as AlphaFold3 (Abramson et al., 2024a), Chai-1 (Chai Discovery et al., 2024), Boltz (Wohlwend et al., 2024; Passaro et al., 2025), Protenix (Protenix Team et al., 2026), SeedFold (Zhou et al., 2025), IntFold (The IntFold Team et al., 2025), NeuralPLexer3 (Qiao et al., 2024), SimpleFold (Wang et al., 2025a), OpenFold3 (The OpenFold3 Team, 2025), and HelixFold3 (Liu et al., 2024) blur the boundary between folding and docking by jointly modeling biomolecular complexes at atomic resolution. Compared with rigid-body docking, co-folding models can better capture global binding topology, protein–ligand contacts, side-chain rearrangements, and local atomic geometry. They define the high-fidelity direction of modern molecular docking.

Yet this accuracy comes with a deployment cost. Co-folding models typically rely on large pair representations, deep attention blocks, and iterative generative refinement. These mechanisms improve structural fidelity, but they also increase latency and memory cost. This leaves an accuracy–latency gap: co-folding models have advanced the pose-accuracy frontier, but remain difficult to deploy at screening scale or inside repeatedly invoked agentic workflows.

### 2.2. Few-Step Generative Dynamics

Modern co-folding models often rely on diffusion-style generative modeling to produce molecular coordinates. Diffusion probabilistic models define a forward corruption process and learn a reverse denoising trajectory (Song et al., 2020). EDM-style formulations (Karras et al., 2022) are widely used in structure generation because they provide stable training and strong generative fidelity. Their limitation is inference cost. Accurate sampling usually requires multiple denoising steps, which directly limits throughput.

Several lines of work aim to reduce this cost. Flow Matching and Rectified Flow learn velocity fields that transport noise to data along straighter probability paths (Lipman et al., 2022; Esser et al., 2024). A key insight is that straighter trajectories can be discretized with fewer solver steps. Consistency Models further reduce sampling cost by enforcing predictions along the same probability-flow trajectory to agree (Song et al., 2023; Song & Dhariwal, 2023). Recent formulations such as MeanFlow connect trajectory straightening and consistency learning under a unified view of few-step generation (Geng et al., 2025).

Most of these developments were established in general image, video, or latent generative modeling. Their implications for molecular docking are less direct. Molecular structures require both global pose correctness and local atomic fidelity. Errors introduced by coarse discretization can disrupt binding geometry, side-chain packing, or interface contacts. Thus, low-NFE molecular generation is not only a question of fewer solver evaluations. It also depends on the geometry of the probability path. The role of trajectory geometry under controlled few-step molecular sampling remains underexplored.

### 2.3. Efficient Co-folding Primitives and Quantization

The deployment of foundation models is increasingly constrained by memory bandwidth and inference latency. Quantization is a standard approach to reduce these costs. Post-Training Quantization (PTQ) compresses a trained model without full retraining (Yao et al., 2022), while Quantization-Aware Training (QAT) exposes the model to quantization noise during training (Nagel et al., 2021). Both approaches

map high-precision weights or activations to lower-bit representations, often through affine quantization. For large Transformers, this can reduce memory traffic and exploit efficient low-precision hardware.

However, co-folding models are not generic Transformers. They propagate geometry-sensitive information through residual streams while performing expensive transformations inside pair, token, and diffusion blocks. Uniformly quantizing all pathways may disturb the numerical stability needed for molecular geometry. Keeping the entire model in high precision avoids this risk, but leaves the main memory and latency bottlenecks unresolved.

This creates a need for structured precision strategies. For molecular docking, efficiency cannot be pursued by compression alone. The precision pathway that carries structural information must be protected, while compute-heavy transformations should be accelerated. This distinction is especially important for co-folding models, where deployment efficiency and structural fidelity are tightly coupled.

## 3. Preliminaries

### 3.1. Diffusion and Transport as Gaussian Interpolation

We view the diffusion and transport parameterizations considered in this work through a common Gaussian interpolation (Song et al., 2020; Karras et al., 2022; Lipman et al., 2022; Esser et al., 2024). Let $\mathbf{x}_0 \in \mathbb{R}^{N \times 3}$ denote a clean molecular structure with $N$ atoms, and let $\boldsymbol{\epsilon} \sim \mathcal{N}(\mathbf{0}, \mathbf{I})$ denote standard Gaussian noise of the same shape. A noisy structure is written as

$$\mathbf{x}_t = \alpha(t)\mathbf{x}_0 + \sigma(t)\boldsymbol{\epsilon}. \tag{1}$$

Equivalently, the perturbation kernel is

$$q(\mathbf{x}_t|\mathbf{x}_0) = \mathcal{N}(\mathbf{x}_t; \alpha(t)\mathbf{x}_0, \sigma^2(t)\mathbf{I}). \tag{2}$$

Here $\alpha(t)$ and $\sigma(t)$ are the signal and noise scales. For clarity, we present the normalized-coordinate case; in implementation, coordinates are centered and scaled consistently. When $\alpha(t) > 0$, the square root of noise-to-signal ratio (NSR) is

$$\sqrt{\text{NSR}} = \lambda(t) = \frac{\sigma(t)}{\alpha(t)}. \tag{3}$$

The native time $t$ is schedule-dependent, while $\lambda$ provides a schedule-independent coordinate for relative corruption.

**EDM denoising.** The EDM or variance-exploding parameterization sets

$$\alpha_{\text{EDM}}(t) = 1, \qquad \sigma_{\text{EDM}}(t) = t. \tag{4}$$

Therefore,

$$\mathbf{x}_t = \mathbf{x}_0 + t\boldsymbol{\epsilon}, \qquad \lambda_{\text{EDM}}(t) = t. \tag{5}$$

With unit data variance, the EDM preconditioned denoiser is

$$D_\theta(\mathbf{x}_t, t) = c_{\text{skip}}(t)\mathbf{x}_t + c_{\text{out}}(t)F_\theta(\mathbf{x}_t, t). \tag{6}$$

The preconditioning coefficients are

$$c_{\text{skip}}(t) = \frac{1}{1+t^2}, \qquad c_{\text{out}}(t) = \frac{t}{\sqrt{1+t^2}}. \tag{7}$$

If $D_\theta$ predicts $\mathbf{x}_0$, the raw target of $F_\theta$ is

$$F_{\text{EDM}} = \frac{\lambda}{\sqrt{1+\lambda^2}}\mathbf{x}_0 - \frac{1}{\sqrt{1+\lambda^2}}\boldsymbol{\epsilon}. \tag{8}$$

Thus, EDM asks the network to learn a noise-level-dependent denoising projection.

**Linear transport.** The Linear parameterization sets

$$\alpha_{\text{lin}}(t) = 1 - t, \qquad \sigma_{\text{lin}}(t) = t, \qquad t \in [0, 1). \tag{9}$$

It follows that

$$\mathbf{x}_t = (1 - t)\mathbf{x}_0 + t\boldsymbol{\epsilon}. \tag{10}$$

The corresponding NSR and inverse map are

$$\lambda_{\text{lin}}(t) = \frac{t}{1-t}, \qquad t_{\text{lin}}(\lambda) = \frac{\lambda}{1+\lambda}. \tag{11}$$

The natural target is the pairwise velocity

$$\mathbf{v}_{\text{lin}} = \frac{d\mathbf{x}_t}{dt} = \boldsymbol{\epsilon} - \mathbf{x}_0. \tag{12}$$

Given a predicted velocity $\mathbf{v}_\theta(\mathbf{x}_t, t)$, the clean endpoint is

$$\hat{\mathbf{x}}_0 = \mathbf{x}_t - t\mathbf{v}_\theta(\mathbf{x}_t, t). \tag{13}$$

For $t > 0$, velocity matching is equivalent to a time-weighted reconstruction objective. Define

$$\mathbf{e}_v = \mathbf{v}_\theta - (\boldsymbol{\epsilon} - \mathbf{x}_0). \tag{14}$$

Define the induced reconstruction error as

$$\mathbf{e}_x = \mathbf{x}_0 - (\mathbf{x}_t - t\mathbf{v}_\theta). \tag{15}$$

Then

$$\|\mathbf{e}_v\|_2^2 = \frac{1}{t^2}\|\mathbf{e}_x\|_2^2. \tag{16}$$

**Root NSR-matched comparison.** A common root NSR coordinate lets us compare different parameterizations under the same relative corruption level. The normalized noisy input is

$$\mathbf{z}_\lambda = \frac{\mathbf{x}_t}{\sqrt{\alpha^2(t) + \sigma^2(t)}}. \tag{17}$$

Using $\lambda(t) = \sigma(t)/\alpha(t)$, this becomes

$$\mathbf{z}_\lambda = \frac{\mathbf{x}_0 + \lambda\boldsymbol{\epsilon}}{\sqrt{1+\lambda^2}}. \tag{18}$$

After root NSR matching and input normalization, EDM-style denoising and Linear transport can expose the network to the same normalized noisy structure $(\mathbf{z}_\lambda, \lambda)$. What changes is the generative question: EDM predicts a time-dependent denoising projection, while Linear predicts a time-independent pairwise transport target, though the model still conditions on $\lambda$.

VP or cosine-style schedules fit the same root NSR view. For example,

$$\mathbf{x}_s^{\text{VP}} = \cos(s)\mathbf{x}_0 + \sin(s)\boldsymbol{\epsilon}. \tag{19}$$

Its root NSR is

$$\lambda_{\text{VP}}(s) = \tan(s). \tag{20}$$

Thus VP/cosine-style schedules are comparable to EDM-style denoising under root NSR matching up to a global scale. By contrast, Linear differs not only in raw interpolation, but also in target parameterization and update rule.

**Deterministic updates.** For deterministic samplers, both parameterizations can be viewed as recovering endpoint estimates and re-encoding them at a lower noise level. For EDM, given $\mathbf{x}_t = \mathbf{x}_0 + t\boldsymbol{\epsilon}$ and a clean estimate $\hat{\mathbf{x}}_0$, the noise estimate is

$$\hat{\boldsymbol{\epsilon}} = \frac{\mathbf{x}_t - \hat{\mathbf{x}}_0}{t}, \qquad t > 0. \tag{21}$$

The lower-noise state is then

$$\mathbf{x}_r = \hat{\mathbf{x}}_0 + r\hat{\boldsymbol{\epsilon}}. \tag{22}$$

Equivalently,

$$\mathbf{x}_r = \mathbf{x}_t + \frac{r-t}{t}\left(\mathbf{x}_t - \hat{\mathbf{x}}_0\right). \tag{23}$$

For Linear, given a velocity estimate $\hat{\mathbf{v}}$, the clean and noise endpoints are

$$\hat{\mathbf{x}}_0 = \mathbf{x}_t - t\hat{\mathbf{v}}. \tag{24}$$

The corresponding noise endpoint is

$$\hat{\boldsymbol{\epsilon}} = \mathbf{x}_t + (1-t)\hat{\mathbf{v}}. \tag{25}$$

Re-encoding at $r < t$ gives

$$\mathbf{x}_r = \mathbf{x}_t + (r-t)\hat{\mathbf{v}}. \tag{26}$$

Thus, Linear reduces to an Euler step along the learned transport velocity, whereas EDM-style denoising relies on a time-dependent denoising projection. Under very low NFE, this difference can change how discretization errors accumulate.

## 4. Methods: Accuracy–Latency Co-design for Real-Time Docking

Our goal is to make co-folding-based docking accurate enough for pose prediction and fast enough for screening-scale deployment. We decompose inference latency into three dominant sources: representation recycling, coordinate-generation evaluations, and arithmetic cost per evaluation. Accordingly, our design compresses the representation loop, the generative sampling loop, and the precision pathway.

### 4.1. Streamlined Co-folding Backbone

A co-folding model contains two computational trunks. The **representation trunk**, such as an Evoformer- or Pairformer-style module, builds protein–ligand context through pair and token representations. The **coordinate-generation trunk**, often implemented as a DiT-style diffusion or transport module, refines noisy atomic coordinates into molecular structures.

These trunks have different repeated-computation costs. Representation modules may be applied multiple times through recycling, while the coordinate generator is evaluated across denoising or transport steps. We denote the number of recycling passes by $R$, the number of generated candidate conformations by $N_c$, and the number of coordinate-generation evaluations per conformation by $S$. The dominant cost of one docking call can be approximated as

$$\mathcal{C}_{\text{call}} \approx R\,\mathcal{C}_{\text{repr}} + N_c S\,\mathcal{C}_{\text{gen}}. \tag{27}$$

Here, $\mathcal{C}_{\text{repr}}$ and $\mathcal{C}_{\text{gen}}$ are the costs of one representation pass and one coordinate-generation pass. The candidate dimension can be partially batched in practice, but Eq. (27) captures the dominant repeated computation.

We set $R = 1$ and remove auxiliary modules whose empirical benefits are marginal, including explicit coordinate-modulation modules and hand-crafted energy-inspired priors (Appendix D). The resulting backbone keeps the operations most relevant to docking: protein–ligand communication, token-level structural reasoning, and coordinate refinement.

The model supports protocol-dependent structural conditioning. In blind docking settings, it uses ligand identity, receptor structure, and minimal pocket or search-region information when available. In strong-prior settings, it can additionally use pocket-centered or interface-informed context, such as approximate pocket centers and residue-level interface indicators. We report these settings separately so that accuracy comparisons are made under matched input information.

## 4.2. Progressive Consistency Regularization

After reducing representation recycling, iterative coordinate generation becomes the main algorithmic bottleneck. We introduce **Progressive Consistency Regularization (PCR)** to reduce the sampling NFE $S$ while retaining reconstruction supervision.

Let $\mathbf{x}_0 \in \mathbb{R}^{N \times 3}$ be the ground-truth generated structure, and let $\mathbf{x}_t$ be its noisy version at time $t$. We write the clean-structure predictor as

$$\hat{\mathbf{x}}_0 = g_\theta(\mathbf{x}_t, t; \mathbf{c}), \tag{28}$$

where $\theta$ denotes model parameters and $\mathbf{c}$ denotes conditioning features determined by the evaluation protocol. The function $g_\theta$ abstracts over parameterization: EDM-style denoising recovers $\hat{\mathbf{x}}_0$ through preconditioning, while Linear transport recovers it from a velocity prediction.

For PCR, adjacent consistency states are constructed from the same clean structure and Gaussian noise. Given $\boldsymbol{\epsilon} \sim \mathcal{N}(\mathbf{0}, \mathbf{I})$ and two adjacent noise levels $t > r$, we form the high-noise point

$$\mathbf{x}_t = \alpha(t)\mathbf{x}_0 + \sigma(t)\boldsymbol{\epsilon}. \tag{29}$$

The lower-noise point is constructed from the same data–noise pair:

$$\mathbf{x}_r = \alpha(r)\mathbf{x}_0 + \sigma(r)\boldsymbol{\epsilon}. \tag{30}$$

Thus, $\mathbf{x}_t$ and $\mathbf{x}_r$ lie on the same Gaussian interpolation path, and no teacher rollout is required to construct the lower-noise point.

For compact notation, define the high-noise clean prediction as

$$\hat{\mathbf{x}}_{0,t} = g_\theta(\mathbf{x}_t, t; \mathbf{c}). \tag{31}$$

The lower-noise stop-gradient target is

$$\bar{\mathbf{x}}_{0,r} = \mathrm{sg}\left[g_{\theta^-}(\mathbf{x}_r, r; \mathbf{c})\right]. \tag{32}$$

In our implementation, $\theta^-$ is an exponential moving average (EMA) of the online parameters $\theta$ and gradients are stopped through the lower-noise branch.

The consistency term is

$$\mathcal{L}_{\mathrm{cons}} = \|\hat{\mathbf{x}}_{0,t} - \bar{\mathbf{x}}_{0,r}\|_2^2, \tag{33}$$

where $\mathcal{L}_{\mathrm{cons}}$ is the l2 distance over generated atoms. The reconstruction anchor is

$$\mathcal{L}_{\mathrm{rec}} = \|\hat{\mathbf{x}}_{0,t} - \mathbf{x}_0\|_2^2. \tag{34}$$

PCR minimizes

$$\mathcal{L}_{\mathrm{PCR}} = \mathbb{E}\left[w_{\mathrm{cons}}(t, r)\mathcal{L}_{\mathrm{cons}} + w_{\mathrm{rec}}(t)\mathcal{L}_{\mathrm{rec}}\right], \tag{35}$$

where the expectation is taken over $\mathbf{x}_0$, $\boldsymbol{\epsilon}$, $t$, and $r$. Here $w_{\mathrm{cons}}$ and $w_{\mathrm{rec}}$ follow the EDM-style weighting convention and reconstruction-anchored consistency setup detailed in Appendix B.8. The reconstruction term anchors the endpoint prediction to the ground-truth structure and prevents consistency tuning from eroding local geometry.

We use a progressive curriculum over the root NSR grid. Let

$$\rho_M = \frac{M_{\max}}{M_{\min}}. \tag{36}$$

Here $k$ indexes the current PCR tuning step and $K_{\mathrm{train}}$ is the total number of tuning steps (100k in our experiments). At step $k$, the number of grid intervals is

$$M(k) = \left\lfloor M_{\min} \rho_M^{k/K_{\mathrm{train}}} \right\rfloor. \tag{37}$$

In our experiments, we set $M_{\min} = 10$ and $M_{\max} = 1000$. The discrete noise levels at each step follow a Karras schedule (Karras et al., 2022), producing a sequence

$$\lambda_{M(k)} > \lambda_{M(k)-1} > \cdots > \lambda_0. \tag{38}$$

For a sampled index $i$, $\lambda_i$ and $\lambda_{i-1}$ define adjacent high- and low-noise levels, which are mapped to schedule times $t$ and $r$. As $M(k)$ increases, adjacent intervals become smaller, moving the objective from coarse trajectory compression toward local consistency.

## 4.3. Residual-Safe Quantization

PCR reduces the number of coordinate-generation evaluations. Residual-Safe Quantization reduces the cost of each evaluation. Co-folding blocks contain two numerical pathways: residual streams that carry accumulated structural information, and inner linear transformations that dominate matrix-multiplication cost. We keep residual streams and geometry-sensitive operations in BF16, while quantizing selected compute-heavy linear transformations inside Pairformer and token-level DiT blocks.

For a residual block, the high-precision computation is

$$\mathbf{y} = \mathbf{x} + F(\mathbf{x}; \mathbf{W}). \tag{39}$$

Residual-Safe Quantization replaces selected linear operations inside $F$ by quantized matrix multiplication. Let the quantized branch output be

$$\mathbf{h} = \mathrm{QuantOp}\left(\mathbf{x}_{\mathrm{BF16}}, \mathbf{W}_{\mathrm{INT8}}; s_x, s_w\right). \tag{40}$$

The residual update is performed in BF16:

$$\mathbf{y}_{\mathrm{BF16}} = \mathbf{x}_{\mathrm{BF16}} + \mathrm{DeQuant}(\mathbf{h}). \tag{41}$$

Here, $\mathbf{W}_{\mathrm{INT8}}$ denotes quantized weights, $s_w$ denotes per-channel weight scales, $s_x$ denotes activation scales used

for W8A8, and $\mathrm{DeQuant}(\cdot)$ maps the low-precision branch output back to BF16 before residual accumulation. Thus, quantization noise is confined to the local transformation branch, while the residual stream remains in BF16.

We use post-training quantization without quantization-aware retraining. For W8A16, selected linear weights are quantized to 8-bit integers and activations remain in BF16. For W8A8, both weights and activations are quantized inside selected linear operations, with activation scales estimated from a validation calibration set before deployment. Attention softmax, normalization layers, triangle multiplication, triangle attention, input embedders, and final structure heads remain in BF16. This separates the geometry-sensitive residual pathway from the throughput-critical matrix multiplication pathway.

Together, representation-loop reduction, PCR, and Residual-Safe Quantization address the three main latency sources considered in this work: recycling passes, sampling evaluations, and arithmetic cost per evaluation.

## 5. Experimental Analysis

Our experiments evaluate whether high-fidelity co-folding docking can be compressed into a screening-scale structural primitive without losing pose accuracy. We organize the analysis around four questions. First, does the diffusion-pretrained backbone provide a strong accuracy foundation under a specified information protocol? Second, how should this accuracy be interpreted when different docking methods use different levels of structural prior? Third, how do EDM and Linear parameterizations behave under low-NFE sampling, and does PCR improve few-step stability? Fourth, can Residual-Safe Quantization reduce inference cost while preserving docking quality?

### 5.1. Evaluation Protocol and Metrics

**Data split.** We train and evaluate on curated protein–ligand complexes derived from PDBBind2020 (Wang et al., 2004b). After filtering low-quality entries and resolving chain-level inconsistencies, the curated PDBBind2020 data are split into approximately 18k training complexes, 1,024 hold-out validation complexes, and 428 held-out test complexes. The test set is not used for training, validation, model selection, PCR tuning, calibration, or quantization. Unless otherwise stated, benchmark accuracy results are reported on the 428-complex test set.

**Information-level-matched evaluation.** Docking methods differ in the amount of structural information available at inference time. Blind docking methods operate with minimal pocket information and must search a broad pose space. Pocket-specified or surface-informed methods re-

ceive stronger spatial cues. Interface-informed methods may further use residue-level or contact-like priors that substantially constrain the docking problem. To avoid conflating model quality with input-information strength, we interpret docking results under an information-level-matched view: blind docking settings should be compared with blind generative docking methods, while strong-prior settings should be compared with surface- or interface-informed baselines.

**Docking success rate.** We evaluate global pose quality using Docking Success Rate (SR). For benchmark accuracy evaluation, we follow the common sample-and-rank protocol used by modern co-folding and generative docking models. For each target, we sample $N = 25$ candidate complexes and select the top-5 predictions according to the model confidence score. A target is counted as successfully docked if at least one of the top-5 predictions achieves ligand RMSD $< 2.0$ Å after pocket alignment. This top-5 SR evaluates whether the model can place at least one high-quality pose among a standard candidate set.

**Pocket side-chain accuracy.** We also report pocket side-chain torsion error, denoted as $\chi_{1-4}$-MAE. Pocket residues are defined as protein residues with any heavy atom within 10 Å of the reference ligand. Torsion errors are measured in degrees using periodic angular distance. Residues or torsion angles without a valid $\chi$ angle are not imputed and are excluded from the average. The final $\chi_{1-4}$-MAE is averaged over valid $\chi_1 - \chi_4$ torsions in the pocket and over the selected top-5 predictions.

**Accuracy and latency protocols.** Table 1 evaluates diffusion-pretrained models before PCR or quantized deployment. Figure 2 evaluates low-NFE sampling, PCR tuning, and precision settings. Latency experiments use a separate deployment-oriented protocol in Section 5.5. The accuracy protocol uses 25 sampled candidates to match standard sample-and-rank evaluation, while the latency protocol generates 5 conformations to measure the cost of a practical screening-time call. Thus, accuracy and latency answer complementary questions: the former measures pose quality under sample-and-rank evaluation, while the latter isolates runtime under a fixed input-shape and output-count setting.

### 5.2. Diffusion Pre-training Establishes a Strong Accuracy Baseline

We first evaluate whether the streamlined Pairformer–DiT backbone is accurate before applying PCR or quantized deployment. Table 1 compares our diffusion-pretrained models with classical search methods, regression-based docking models, and recent generative or refinement baselines on the PDBBind2020 test set. Because docking methods differ in the structural information available at inference time,

the table should be read together with the information-level analysis in Appendix D.2.

With a standard 20-NFE sampling budget, our Linear and EDM variants achieve docking SRs of 83.18% and 82.48%, respectively, under the interface-informed protocol. These results place the streamlined co-folding backbone at the high-accuracy frontier before few-step consistency tuning or Residual-Safe Quantization is applied. They also motivate the subsequent analysis: high accuracy is achievable, but its interpretation depends on the input-information regime.

*Table 1.* **Comparative Analysis on the PDBBind2020 Test Set.** We report Docking Success Rate (SR, %), defined as the percentage of targets with at least one top-5 prediction achieving ligand RMSD $< 2.0$ Å after pocket alignment. The table follows the evaluation protocols reported for each method, and different methods may use different input-information levels. Our diffusion-pretrained models are evaluated under an interface-informed strong-prior protocol; therefore, the highest-accuracy numbers should be interpreted as interface-informed docking performance rather than blind docking performance.

| Method | NFE | SR (%) |
|---|---|---|
| *Classical & Regression* | | |
| Vina | – | 52.30 |
| EquiBind | 1 | 2.58 |
| TANKBind | 1 | 14.95 |
| *Generative & Refinement* | | |
| DiffDock | 20 | 47.66 |
| DiffDock-L | 20 | 50.00 |
| Uni-Mol Docking V2 | – | 61.21 |
| Interformer | – | 77.00 |
| SurfDock (minimized) | 20 | 81.07 |
| **Ours (Linear)** | **20** | **83.18** |
| **Ours (EDM)** | **20** | **82.48** |

### 5.3. Input Information Level and Evaluation Scope

Docking accuracy depends strongly on the amount of structural information available at inference time. Blind docking evaluates global search under minimal spatial guidance, while pocket-specified, surface-informed, and interface-informed settings evaluate increasingly constrained pose generation or refinement. A systematic analysis of this information-conditioning hierarchy is provided in Appendix D.2.

Our highest-accuracy result in Table 1 uses the interface-informed protocol and should not be interpreted as blind docking performance.

### 5.4. Trajectory Geometry and PCR Enable Few-Step Docking

We next analyze inference dynamics under varying NFEs. This experiment tests the central claim of our method: few-step molecular generation is not only a matter of reducing

solver steps, but also a matter of trajectory geometry and consistency.

Figure 2 summarizes the behavior of EDM and Linear parameterizations before and after PCR, together with the effect of Residual-Safe Quantization.

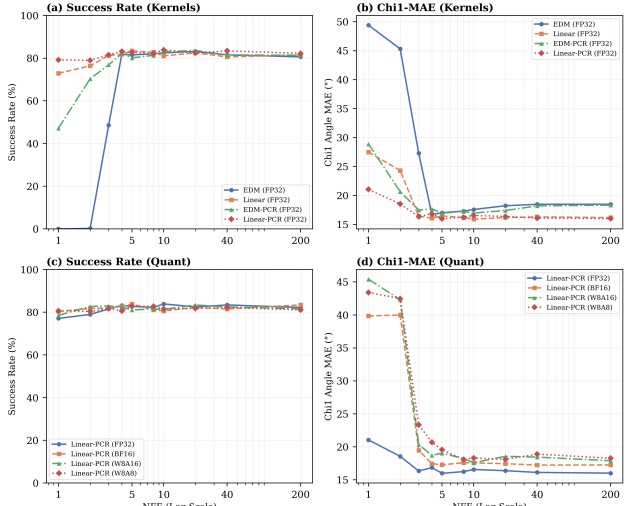

*Figure 2.* **Ablation Study on Inference Dynamics and Precision.** (a) Docking SR of pretrained and PCR-tuned models across varying NFEs. (b) Pocket-residue $\chi_{1-4}$-MAE for pretrained and PCR-tuned models. (c) Docking SR under different precision settings: FP32, BF16, W8A16, and W8A8. (d) Pocket-residue $\chi_{1-4}$-MAE under different precision settings.

**Low-NFE behavior.** For NFE $\geq 4$, EDM and Linear models achieve comparable docking SR, with only modest differences in pocket torsion accuracy. The difference becomes clear in the ultra-low-NFE regime. At 2 NFE, the standard EDM model degrades sharply, while the Linear model retains substantially better docking fidelity. PCR improves few-step stability for EDM, but Linear-PCR remains stronger in the lowest-NFE setting.

This behavior is consistent with the trajectory-geometry view developed in Section 3.1. The Linear parameterization uses a time-independent pairwise velocity target and a simple transport update. By contrast, EDM relies on a time-dependent denoising projection from which clean and noise endpoints are recovered. When the inference grid becomes extremely coarse, this difference affects how discretization error accumulates. PCR further reduces this error by enforcing cross-time consistency while retaining reconstruction supervision.

**Residual-Safe Quantization.** Panels (c) and (d) evaluate whether Residual-Safe Quantization changes docking quality. The global docking SR remains stable under W8A16 and W8A8, suggesting that global pose prediction is relatively robust when quantization is restricted to selected

linear transformations and residual accumulation remains in BF16. The local torsion metric is more sensitive. Under more aggressive quantization, $\chi_{1-4}$-MAE increases, indicating that fine-grained side-chain geometry is more sensitive to numerical perturbations than global ligand placement.

Together, these results identify the 4–10 NFE range as a practical operating window. Within this range, Linear-PCR provides strong docking accuracy while substantially reducing sampling cost. Residual-Safe Quantization further reduces per-step compute, making this regime suitable for screening-scale inference.

### 5.5. End-to-End Inference Latency

Finally, we evaluate whether the proposed design translates into real inference speed. Latency is measured on a single NVIDIA H20 GPU. All methods are evaluated under the same input-shape and output-count setting: generating 5 candidate conformations for a representative 256-token complex. We report synchronized GPU time averaged over 100 runs after warmup. The timed region excludes feature preprocessing, confidence ranking, and compilation overhead. For external baselines such as AlphaFold3 and Chai-1, we use their official implementations under the same hardware and candidate-generation setting whenever supported.

As shown in Figure 3, existing co-folding models remain expensive because they rely on large pair/token representations, recycling-style refinement, and iterative coordinate generation. Specialized docking models reduce part of this cost, but remain limited by their own sampling or implementation bottlenecks. Our model combines three reductions: representation recycling is reduced to a single pass, PCR reduces the sampling NFE to 5, and W8A8 Residual-Safe Quantization with JIT compilation accelerates compute-heavy linear operations. This yields 0.17 s GPU latency for generating 5 conformations, corresponding to a $> 300\times$ speedup over AlphaFold3 under the benchmarked input-shape setting.

These experiments support the main conclusion: high-fidelity co-folding docking can be compressed into a sub-second structural primitive.

## 6. Contributions

This work reframes molecular docking as a real-time structural primitive for screening-scale and agent-centered discovery. Instead of treating accuracy and efficiency as separate objectives, we co-design the model architecture, generative trajectory, and deployment precision. Our contributions are:

1. **Docking as a sub-second structural primitive.** We develop a streamlined co-folding framework that pre-

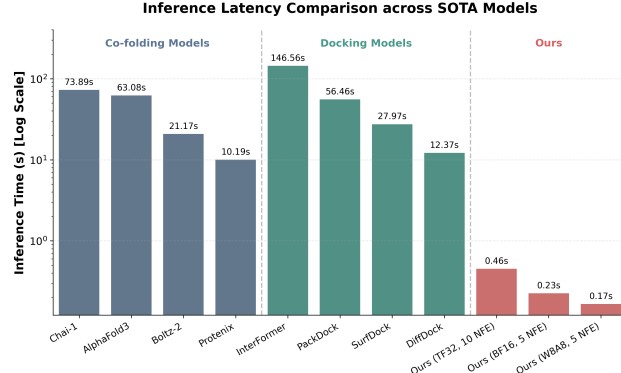

*Figure 3.* **Inference Latency Benchmark.** GPU time on a single NVIDIA H20 GPU to generate 5 conformations for a representative 256-token complex. The timed region excludes feature preprocessing, confidence ranking, and compilation overhead, and reports synchronized GPU time averaged over 100 runs after warmup. By combining representation-loop reduction, PCR with NFE=5, W8A8 Residual-Safe Quantization, and JIT compilation, our model reaches 0.17 s latency, corresponding to a $> 300\times$ speedup over AlphaFold3 under the benchmarked input-shape setting.

serves the core computation needed for protein–ligand reasoning while reducing redundant inference loops. The model achieves strong docking accuracy under matched information protocols while remaining suitable for low-latency deployment.

2. **Trajectory-aware few-step generation.** We show that low-NFE molecular generation depends not only on solver count, but also on trajectory geometry. This motivates **Progressive Consistency Regularization (PCR)**, which compresses diffusion-based docking into reliable few-step inference through reconstruction-anchored consistency tuning.

3. **Residual-Safe Quantization for co-folding inference.** We introduce a mixed-precision deployment strategy that keeps residual streams and geometry-sensitive operations in BF16 while quantizing selected compute-heavy linear transformations. This separates the structural precision path from the throughput-critical computation path.

4. **An accuracy–latency frontier for real-time docking.** Combining representation-loop reduction, PCR, and Residual-Safe Quantization, our system reaches sub-second inference while preserving high docking accuracy. It establishes molecular docking as a callable structural module for high-throughput screening and agent-centered drug discovery.

We have released part of the implementation at https://github.com/KexinZhangResearch/PhysDock to facilitate reproducibility and future research.

## Acknowledgements

We thank the anonymous reviewers for their constructive feedback and insightful suggestions. We especially appreciate their recognition of the industrial potential of low-latency molecular docking, which helped us better clarify the deployment motivation, evaluation protocols, and practical role of our model as a real-time structural primitive for drug discovery.

This work was supported in part by the National Natural Science Foundation of China under Grant W2431046, Central Guided Local Science and Technology Foundation of China YDZX20253100001001, ShanghaiTech AI Initiative (Grant No. AI2026A02), National Key R&D Program of China 2025YFA1309603, and by MoE Key Lab of Intelligent Perceptionand Human-Machine Collaboration (ShanghaiTech University), the Shanghai Frontiers Science Center of Human-centered Artificial Intelligence, and HPC Platform of ShanghaiTech University.

## Impact Statement

This work aims to make high-fidelity molecular docking fast enough to serve as a repeatedly callable structural primitive. Its main scientific impact is to shift co-folding-based docking from an offline generative simulator toward a low-latency component for structure-based discovery. In virtual screening, this matters because the practical value of a docking model depends not only on pose accuracy for individual complexes, but also on how efficiently it can evaluate large candidate libraries.

For drug discovery, the proposed framework can increase the practical throughput of deep-learning-based docking. By reducing both the number of generative evaluations and the cost of each evaluation, it makes high-fidelity structural feedback more compatible with screening-scale use cases where faster but less expressive approximations are often preferred. The model is not a replacement for experimental validation. Rather, it provides faster computational feedback for prioritizing candidates before expensive wet-lab assays.

For agent-centered scientific discovery, sub-second docking enables a more interactive workflow. A scientific agent can invoke docking repeatedly while ranking candidates, revising molecular hypotheses, or exploring chemical space. In this setting, latency is not merely a systems metric; it determines whether structural prediction can remain inside an active reasoning loop instead of becoming a separate offline batch process.

The efficiency gains may also broaden access to advanced structural modeling. Lower latency and mixed-precision deployment can reduce computational barriers for academic laboratories, non-profit groups, and early-stage biotechnology teams. Such access is beneficial only when paired with transparent benchmarking, careful validation, and clear reporting of protocol assumptions, including the amount of structural prior used at inference time.

This work also raises dual-use considerations. Although the method does not itself generate new molecules or biological sequences, faster docking can accelerate the evaluation and prioritization of molecular candidates, including candidates intended for harmful use. Responsible deployment should therefore include appropriate access controls, usage monitoring, safety filters for restricted chemical or biological targets, and experimental validation before downstream decisions. The broader governance challenge is not only model capability, but how high-throughput screening systems are integrated, audited, and used.

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

# A. Unifying Diffusion and Transport Parameterizations

This appendix formalizes the diffusion and transport parameterizations used in our controlled comparison. The purpose is not to claim that EDM, VE, VP, cosine, and Linear formulations are identical algorithms. They differ in native variables, raw coordinate scales, prediction targets, and deterministic update rules. Our claim is narrower: after expressing corruption by the noise-to-signal ratio (NSR) and normalizing the noisy input, several schedule-level differences can be aligned. This allows us to compare target parameterizations and update geometries under the same relative corruption coordinate.

Throughout this appendix, let $\mathbf{x}_0 \in \mathbb{R}^{N \times 3}$ denote the clean molecular coordinates of the generated atoms, and let $\boldsymbol{\epsilon} \sim \mathcal{N}(\mathbf{0}, \mathbf{I})$ denote Gaussian noise of the same shape. All norms are understood over the generated coordinate entries; masks are omitted for clarity.

## A.1. Common Gaussian Form and NSR

A broad class of noising processes can be written as

$$\mathbf{x}_s^{\mathcal{S}} = \alpha_{\mathcal{S}}(s)\mathbf{x}_0 + \sigma_{\mathcal{S}}(s)\boldsymbol{\epsilon}, \tag{42}$$

where $\mathcal{S}$ denotes the parameterization, $s$ is its native variable, $\alpha_{\mathcal{S}}(s)$ is the signal scale, and $\sigma_{\mathcal{S}}(s)$ is the noise scale. The perturbation kernel is

$$q_{\mathcal{S}}(\mathbf{x}_s^{\mathcal{S}} \mid \mathbf{x}_0) = \mathcal{N}\left(\mathbf{x}_s^{\mathcal{S}}; \alpha_{\mathcal{S}}(s)\mathbf{x}_0, \sigma_{\mathcal{S}}^2(s)\mathbf{I}\right). \tag{43}$$

When $\alpha_{\mathcal{S}}(s) > 0$, the amplitude-level root noise-to-signal ratio is

$$\lambda_{\mathcal{S}}(s) = \frac{\sigma_{\mathcal{S}}(s)}{\alpha_{\mathcal{S}}(s)}. \tag{44}$$

The native variable $s$ is schedule-dependent, while $\lambda$ is a schedule-independent coordinate for relative corruption.

Using $\lambda = \sigma_{\mathcal{S}}(s)/\alpha_{\mathcal{S}}(s)$, Eq. (42) can be rewritten as

$$\mathbf{x}_s^{\mathcal{S}} = \alpha_{\mathcal{S}}(s)\left(\mathbf{x}_0 + \lambda\boldsymbol{\epsilon}\right). \tag{45}$$

We define the NSR-normalized input by removing the total signal–noise scale:

$$\mathbf{z}_\lambda = \frac{\mathbf{x}_s^{\mathcal{S}}}{\sqrt{\alpha_{\mathcal{S}}^2(s) + \sigma_{\mathcal{S}}^2(s)}}. \tag{46}$$

Substituting Eq. (45) into Eq. (46) gives

$$\mathbf{z}_\lambda = \frac{\mathbf{x}_0 + \lambda\boldsymbol{\epsilon}}{\sqrt{1 + \lambda^2}}. \tag{47}$$

Thus, different native schedules can expose the network to the same normalized noisy structure once they are matched by $\lambda$. This statement concerns the input coordinate. It does not imply that the prediction targets or sampling updates are the same.

## A.2. Denoising Parameterizations

The denoising parameterizations considered here can be written as scaled versions of the same canonical noisy direction:

$$\mathbf{x}_\lambda^{\text{den}} = a(\lambda)\left(\mathbf{x}_0 + \lambda\boldsymbol{\epsilon}\right), \qquad a(\lambda) > 0. \tag{48}$$

The scale $a(\lambda)$ depends on the chosen native schedule. After normalization, all of these forms reduce to Eq. (47).

**VE / EDM form.** For the VE or EDM-style raw coordinate,

$$\mathbf{x}_t = \mathbf{x}_0 + t\boldsymbol{\epsilon}, \tag{49}$$

we have

$$\lambda = t, \qquad a(\lambda) = 1. \tag{50}$$

**Normalized EDM / VP-like form.** For the unit-total-scale form,

$$\mathbf{x}_t = \frac{\mathbf{x}_0 + t\boldsymbol{\epsilon}}{\sqrt{1 + t^2}}, \tag{51}$$

we have

$$\lambda = t, \qquad a(\lambda) = \frac{1}{\sqrt{1 + \lambda^2}}. \tag{52}$$

**VP noise-amplitude form.** For the VP noise-amplitude parameterization,

$$\mathbf{x}_t = \sqrt{1 - t^2}\mathbf{x}_0 + t\boldsymbol{\epsilon}, \qquad t \in [0, 1), \tag{53}$$

the root NSR is

$$\lambda = \frac{t}{\sqrt{1 - t^2}}. \tag{54}$$

Solving for the native noise amplitude gives

$$t = \frac{\lambda}{\sqrt{1 + \lambda^2}}, \qquad \sqrt{1 - t^2} = \frac{1}{\sqrt{1 + \lambda^2}}. \tag{55}$$

Therefore,

$$a(\lambda) = \frac{1}{\sqrt{1 + \lambda^2}}. \tag{56}$$

Here "VP" refers to this noise-amplitude form, not to a particular continuous-time VP SDE schedule.

**Cosine-style form.** For the cosine-angle parameterization,

$$\mathbf{x}_\tau = \cos(\tau)\mathbf{x}_0 + \sin(\tau)\boldsymbol{\epsilon}, \qquad \tau \in [0, \pi/2), \tag{57}$$

the root NSR relation is

$$\lambda = \tan(\tau), \qquad \tau = \arctan(\lambda). \tag{58}$$

Thus,

$$\cos(\tau) = \frac{1}{\sqrt{1 + \lambda^2}}, \qquad \sin(\tau) = \frac{\lambda}{\sqrt{1 + \lambda^2}}, \tag{59}$$

and again

$$a(\lambda) = \frac{1}{\sqrt{1 + \lambda^2}}. \tag{60}$$

### A.3. Canonical Denoising Target and Recovery

For denoising parameterizations, the normalized noisy input is

$$\mathbf{z}_\lambda = \frac{\mathbf{x}_0 + \lambda\boldsymbol{\epsilon}}{\sqrt{1 + \lambda^2}}. \tag{61}$$

We use the orthogonal denoising target

$$\mathbf{d}_\lambda^{\text{den}} = \frac{\lambda\mathbf{x}_0 - \boldsymbol{\epsilon}}{\sqrt{1 + \lambda^2}}. \tag{62}$$

Together, Eqs. (61) and (62) define the transform

$$\begin{bmatrix} \mathbf{z}_\lambda \\ \mathbf{d}_\lambda^{\text{den}} \end{bmatrix} = \frac{1}{\sqrt{1 + \lambda^2}} \begin{bmatrix} 1 & \lambda \\ \lambda & -1 \end{bmatrix} \begin{bmatrix} \mathbf{x}_0 \\ \boldsymbol{\epsilon} \end{bmatrix}. \tag{63}$$

The matrix in Eq. (63) is self-inverse after normalization. Therefore, given a prediction $\mathbf{d}_\theta \approx \mathbf{d}_\lambda^{\text{den}}$, the endpoints are recovered by

$$\hat{\mathbf{x}}_0 = \frac{\mathbf{z}_\lambda + \lambda\mathbf{d}_\theta}{\sqrt{1 + \lambda^2}}, \tag{64}$$

and

$$\hat{\boldsymbol{\epsilon}} = \frac{\lambda \mathbf{z}_\lambda - \mathbf{d}_\theta}{\sqrt{1 + \lambda^2}}. \tag{65}$$

If the raw state is $\mathbf{x}_\lambda^{\mathrm{den}} = a(\lambda)(\mathbf{x}_0 + \lambda \boldsymbol{\epsilon})$, then the normalized input used in Eqs. (64)–(65) is

$$\mathbf{z}_\lambda = \frac{\mathbf{x}_\lambda^{\mathrm{den}}}{a(\lambda)\sqrt{1 + \lambda^2}}. \tag{66}$$

The same target can be expressed in native variables. For VE / EDM and the normalized EDM / VP-like form, $t = \lambda$, so

$$\mathbf{d}_t^{\mathrm{EDM}} = \frac{t\mathbf{x}_0 - \boldsymbol{\epsilon}}{\sqrt{1 + t^2}}. \tag{67}$$

For the VP noise-amplitude form, $t = \lambda/\sqrt{1 + \lambda^2}$, hence

$$\mathbf{d}_t^{\mathrm{VP}} = t\mathbf{x}_0 - \sqrt{1 - t^2}\boldsymbol{\epsilon}. \tag{68}$$

For the cosine-style form, $\lambda = \tan(\tau)$, hence

$$\mathbf{d}_\tau^{\mathrm{cos}} = \sin(\tau)\mathbf{x}_0 - \cos(\tau)\boldsymbol{\epsilon}. \tag{69}$$

**Deterministic denoising update.** A deterministic step from $\lambda$ to a lower noise level $\lambda' < \lambda$ is obtained by recovering the endpoints and re-encoding them:

$$\mathbf{x}_{\lambda'}^{\mathrm{den}} = a(\lambda')\left(\hat{\mathbf{x}}_0 + \lambda'\hat{\boldsymbol{\epsilon}}\right). \tag{70}$$

If the sampler is represented in normalized coordinates, the corresponding next normalized state is

$$\mathbf{z}_{\lambda'} = \frac{\hat{\mathbf{x}}_0 + \lambda'\hat{\boldsymbol{\epsilon}}}{\sqrt{1 + (\lambda')^2}}. \tag{71}$$

Equation (70) gives the VE / EDM, normalized EDM / VP-like, VP noise-amplitude, and cosine-style deterministic updates after applying the appropriate native map.

### A.4. Linear Transport Parameterization

The Linear parameterization uses

$$\mathbf{x}_u^{\mathrm{lin}} = (1 - u)\mathbf{x}_0 + u\boldsymbol{\epsilon}, \qquad u \in [0, 1). \tag{72}$$

Its root NSR is

$$\lambda = \frac{u}{1 - u}, \tag{73}$$

so

$$u = \frac{\lambda}{1 + \lambda}, \qquad 1 - u = \frac{1}{1 + \lambda}. \tag{74}$$

The raw Linear state can therefore be written as

$$\mathbf{x}_u^{\mathrm{lin}} = \frac{1}{1 + \lambda}\left(\mathbf{x}_0 + \lambda\boldsymbol{\epsilon}\right). \tag{75}$$

The total scale of the Linear state is

$$c_u = \sqrt{(1 - u)^2 + u^2} = \frac{\sqrt{1 + \lambda^2}}{1 + \lambda}. \tag{76}$$

Thus the normalized Linear input is

$$\mathbf{z}_\lambda = \frac{\mathbf{x}_u^{\mathrm{lin}}}{c_u} = \frac{\mathbf{x}_0 + \lambda\boldsymbol{\epsilon}}{\sqrt{1 + \lambda^2}}. \tag{77}$$

Therefore, Linear transport can use the same normalized noisy input as the denoising family. The difference lies in the supervised target and the update rule.

The natural pairwise Linear target is the velocity

$$\mathbf{v} = \frac{d\mathbf{x}_u^{\text{lin}}}{du} = \boldsymbol{\epsilon} - \mathbf{x}_0. \tag{78}$$

For a fixed data–noise pair $(\mathbf{x}_0, \boldsymbol{\epsilon})$, this target is independent of $u$ and $\lambda$. The learned velocity field may still depend on the noisy input and the noise level, but the supervised pairwise target itself is time-independent.

Given a prediction $\mathbf{d}_\theta \approx \mathbf{v}$, the endpoint recovery in raw Linear coordinates is

$$\hat{\mathbf{x}}_0 = \mathbf{x}_u^{\text{lin}} - u\mathbf{d}_\theta, \tag{79}$$

and

$$\hat{\boldsymbol{\epsilon}} = \mathbf{x}_u^{\text{lin}} + (1 - u)\mathbf{d}_\theta. \tag{80}$$

If the network input is the normalized coordinate $\mathbf{z}_\lambda$, then $\mathbf{x}_u^{\text{lin}} = c_u\mathbf{z}_\lambda$, and the same recovery can be written as

$$\hat{\mathbf{x}}_0 = c_u\mathbf{z}_\lambda - u\mathbf{d}_\theta, \tag{81}$$

$$\hat{\boldsymbol{\epsilon}} = c_u\mathbf{z}_\lambda + (1 - u)\mathbf{d}_\theta. \tag{82}$$

A deterministic step from $u$ to a lower noise level $u' < u$ is obtained by re-encoding the recovered endpoints:

$$\mathbf{x}_{u'}^{\text{lin}} = (1 - u')\hat{\mathbf{x}}_0 + u'\hat{\boldsymbol{\epsilon}}. \tag{83}$$

Substituting Eqs. (79)–(80) into Eq. (83) gives

$$\mathbf{x}_{u'}^{\text{lin}} = \mathbf{x}_u^{\text{lin}} - (u - u')\mathbf{d}_\theta. \tag{84}$$

Thus the Linear deterministic update is an Euler step along the learned transport velocity. If the sampler is stored in normalized coordinates, the next normalized state is

$$\mathbf{z}_{\lambda'} = \frac{\mathbf{x}_{u'}^{\text{lin}}}{c_{u'}}, \qquad u' = \frac{\lambda'}{1 + \lambda'}. \tag{85}$$

## A.5. Training-Time Sampling and Network Time Conditioning

Let $p_\lambda(\lambda)$ be the training distribution over root NSR. For a parameterization $\mathcal{S}$ with native variable $s$ and monotone map $\lambda = \lambda_\mathcal{S}(s)$, sampling $\lambda \sim p_\lambda$ induces the native-variable density

$$p_\mathcal{S}(s) = p_\lambda\left(\lambda_\mathcal{S}(s)\right)\left|\frac{d\lambda_\mathcal{S}(s)}{ds}\right|. \tag{86}$$

Equivalently, one may sample $\lambda$ first and map it to the native variable:

$$\begin{aligned}
s_{\text{VE/EDM}}(\lambda) &= \lambda, \\
s_{\text{norm}}(\lambda) &= \lambda, \\
s_{\text{VP}}(\lambda) &= \frac{\lambda}{\sqrt{1 + \lambda^2}}, \\
s_{\cos}(\lambda) &= \arctan(\lambda), \\
s_{\text{lin}}(\lambda) &= \frac{\lambda}{1 + \lambda}.
\end{aligned} \tag{87}$$

Here $s_{\text{VE/EDM}}$, $s_{\text{norm}}$, and $s_{\text{VP}}$ correspond to native time $t$; $s_{\cos}$ corresponds to the cosine angle $\tau$; and $s_{\text{lin}}$ corresponds to the Linear interpolation time $u$.

Although the native variables differ across parameterizations, the scalar time input to the network is shared. To avoid confusion with the cosine angle $\tau$, we denote the network time embedding input by $\phi_{\text{net}}$:

$$\phi_{\text{net}} = \frac{1}{4} \ln(\lambda). \tag{88}$$

Thus all schedules use the same schedule-independent root NSR coordinate for network conditioning. Native variables are used only to construct the noisy state and to define the recovery/update map. In practice, we use $\lambda_{\min} > 0$ so that $\ln(\lambda)$ is well defined.

### A.6. Reconstruction-Equivalent Loss Weighting

Let $w_x(\lambda)$ denote the desired endpoint reconstruction weight:

$$\mathcal{L}_x = \mathbb{E}_{\lambda \sim p_\lambda} \left[ w_x(\lambda) \|\hat{\mathbf{x}}_0 - \mathbf{x}_0\|_F^2 \right]. \tag{89}$$

If training is implemented directly on the recovered endpoint $\hat{\mathbf{x}}_0$, the same endpoint weight $w_x(\lambda)$ can be used across parameterizations. If training is instead written as a raw-target regression loss, the raw-target weights must account for the parameterization-specific recovery map.

For the denoising target, Eq. (64) gives

$$\hat{\mathbf{x}}_0 - \mathbf{x}_0 = \frac{\lambda}{\sqrt{1 + \lambda^2}} \left( \mathbf{d}_\theta - \mathbf{d}_\lambda^{\text{den}} \right). \tag{90}$$

Therefore, the raw denoising loss

$$\mathcal{L}_{\text{den}} = \mathbb{E}_{\lambda \sim p_\lambda} \left[ w_{\text{den}}(\lambda) \|\mathbf{d}_\theta - \mathbf{d}_\lambda^{\text{den}}\|_F^2 \right] \tag{91}$$

is reconstruction-equivalent to Eq. (89) when

$$w_{\text{den}}(\lambda) = w_x(\lambda) \frac{\lambda^2}{1 + \lambda^2}. \tag{92}$$

For Linear transport, Eq. (79) gives

$$\hat{\mathbf{x}}_0 - \mathbf{x}_0 = -u \left( \mathbf{d}_\theta - \mathbf{v} \right). \tag{93}$$

Since $u = \lambda/(1 + \lambda)$, the raw velocity loss

$$\mathcal{L}_{\text{lin}} = \mathbb{E}_{\lambda \sim p_\lambda} \left[ w_{\text{lin}}(\lambda) \|\mathbf{d}_\theta - \mathbf{v}\|_F^2 \right] \tag{94}$$

is reconstruction-equivalent to Eq. (89) when

$$w_{\text{lin}}(\lambda) = w_x(\lambda) \left( \frac{\lambda}{1 + \lambda} \right)^2. \tag{95}$$

Thus, matched endpoint weighting does not imply identical raw-target weights. The raw weights differ because the endpoint recovery maps differ.

### A.7. Inference-Time Root NSR Grids

Inference can also be specified in the root NSR coordinate. Let $\lambda_{\max}$ and $\lambda_{\min}$ be the largest and smallest noise levels, and let $K$ be the number of solver intervals. For a monotone warping function $\psi$, define

$$\Delta_\psi = \psi(\lambda_{\max}) - \psi(\lambda_{\min}). \tag{96}$$

An increasing grid from low to high noise is

$$\lambda_i = \psi^{-1} \left( \psi(\lambda_{\min}) + \frac{i}{K} \Delta_\psi \right), \qquad i = 0, \ldots, K. \tag{97}$$

Then $\lambda_0 = \lambda_{\min}$ and $\lambda_K = \lambda_{\max}$. Deterministic sampling proceeds from $i = K$ down to $i = 0$.

A Karras-style grid can be written as

$$\psi(\lambda) = \lambda^{1/\rho}, \qquad \rho > 0. \tag{98}$$

The same root NSR grid is mapped to each native variable using Eq. (87). Therefore, different parameterizations can use the same number of function evaluations and the same sequence of relative corruption levels, even though their native time values differ.

*Table 2.* **Summary of diffusion and transport parameterizations.** All parameterizations can be expressed by the root NSR coordinate $\lambda$ and can use the same normalized input $\mathbf{z}_\lambda$. They differ in native variable, raw state scale, supervised target, and deterministic update.

| Parameterization | Raw noisy state | Native map from $\lambda$ | Supervised target | Deterministic update |
|---|---|---|---|---|
| VE / EDM | $\mathbf{x}_t = \mathbf{x}_0 + t\boldsymbol{\epsilon}$ | $t = \lambda$ | $\dfrac{t\mathbf{x}_0 - \boldsymbol{\epsilon}}{\sqrt{1+t^2}}$ | $\mathbf{x}_{t'} = \hat{\mathbf{x}}_0 + t'\hat{\boldsymbol{\epsilon}}$ |
| Normalized EDM / VP-like | $\mathbf{x}_t = \dfrac{\mathbf{x}_0 + t\boldsymbol{\epsilon}}{\sqrt{1+t^2}}$ | $t = \lambda$ | $\dfrac{t\mathbf{x}_0 - \boldsymbol{\epsilon}}{\sqrt{1+t^2}}$ | $\mathbf{x}_{t'} = \dfrac{\hat{\mathbf{x}}_0 + t'\hat{\boldsymbol{\epsilon}}}{\sqrt{1+(t')^2}}$ |
| VP noise-amplitude | $\mathbf{x}_t = \sqrt{1-t^2}\mathbf{x}_0 + t\boldsymbol{\epsilon}$ | $t = \dfrac{\lambda}{\sqrt{1+\lambda^2}}$ | $t\mathbf{x}_0 - \sqrt{1-t^2}\boldsymbol{\epsilon}$ | $\mathbf{x}_{t'} = \sqrt{1-(t')^2}\hat{\mathbf{x}}_0 + t'\hat{\boldsymbol{\epsilon}}$ |
| Cosine-style | $\mathbf{x}_\tau = \cos(\tau)\mathbf{x}_0 + \sin(\tau)\boldsymbol{\epsilon}$ | $\tau = \arctan(\lambda)$ | $\sin(\tau)\mathbf{x}_0 - \cos(\tau)\boldsymbol{\epsilon}$ | $\mathbf{x}_{\tau'} = \cos(\tau')\hat{\mathbf{x}}_0 + \sin(\tau')\hat{\boldsymbol{\epsilon}}$ |
| Linear | $\mathbf{x}_u = (1-u)\mathbf{x}_0 + u\boldsymbol{\epsilon}$ | $u = \dfrac{\lambda}{1+\lambda}$ | $\mathbf{v} = \boldsymbol{\epsilon} - \mathbf{x}_0$ | $\mathbf{x}_{u'} = \mathbf{x}_u - (u - u')\mathbf{d}_\theta$ |

### A.8. What Is Matched and What Remains Different

The root NSR alignment above matches three things. First, all parameterizations can expose the network to the same normalized noisy coordinate $\mathbf{z}_\lambda$. Second, all parameterizations use the same scalar network time input $\phi_{\text{net}} = \ln(\lambda)/4$. Third, training-time sampling and inference-time discretization can be specified by the same root NSR distribution or grid.

What remains different is the predicted object and the deterministic update geometry. Denoising parameterizations predict the noise-level-dependent projection $\mathbf{d}_\lambda^{\text{den}}$. Linear transport predicts the pairwise velocity $\mathbf{v} = \boldsymbol{\epsilon} - \mathbf{x}_0$, which is independent of $u$ for a fixed data–noise pair. Consequently, denoising samplers update by recovering and re-encoding endpoints along a denoising trajectory, whereas Linear transport updates by an Euler step along a learned velocity field. This distinction is especially important in few-step sampling, where coarse updates make error accumulation strongly dependent on update geometry rather than on the native schedule alone.

Table 2 summarizes the parameterizations used in our comparison.

### A.9. Summary

Root NSR matching is a control device, not an equivalence claim. Under the normalization used in this work, EDM/VE, normalized VP-like, VP noise-amplitude, cosine-style, and Linear parameterizations can share the same corrupted input coordinate, the same network time conditioning, and the same inference grid. The remaining difference is the target parameterization and the update induced by that target. Denoising models predict a noise-level-dependent projection and update by endpoint recovery and re-encoding. Linear transport targets a time-independent pairwise velocity and updates by an Euler step along the learned transport direction. This algebraic distinction provides the basis for our few-step comparison in the main text.

## B. Implementation Details: Featurization, Architecture, and Objectives

This appendix describes the implementation details of our docking model. The main text defines the method at the conceptual level. Here we specify the input features, representation spaces, model topology, generative parameterizations, training objectives, quantization implementation, and inference configuration at a reproducible level. We follow the notation of previous section: $\lambda$ denotes the root NSR, the native interpolation time depends on the parameterization, and $\tau_{\text{net}}$ denotes the scalar time variable fed into the network time encoder.

### B.1. Data Curation and Splits

We train, validate, and test the model on curated protein–ligand complexes derived from PDBBind2020 (Wang et al., 2004a). After filtering low-quality entries and resolving chain-level inconsistencies, the curated data are partitioned into approximately 18k training complexes, 1,024 hold-out validation complexes, and 428 held-out test complexes. The test set is not used for training, validation, model selection, PCR tuning, calibration, or quantization. During training, complexes are sampled uniformly to expose the model to diverse pocket environments, ligand chemotypes, and binding-affinity regimes.

The model is designed for MSA-free docking. It does not use MSA search, MSA row attention, or MSA column attention. Instead, all conditioning information is constructed from atom-level chemistry, token identity, pairwise geometry, ligand

connectivity, protein structural priors, and pocket context.

## B.2. Protocol-Dependent Structural Priors

Docking methods differ in the amount of structural information available at inference time. To make this distinction explicit, we construct pocket and interface features according to the evaluation protocol. In blind docking, spatial prior features are absent or set to their default "not available" values. In noisy-pocket settings, the pocket center is perturbed from the reference pocket center by a controlled random shift. In exact-pocket settings, the pocket center is provided as a strong re-docking prior. In interface-informed strong-prior settings, residue-level interface indicators are additionally provided to match surface- or interface-informed docking baselines.

This design makes structural prior strength an explicit part of the evaluation protocol. The same feature schema is used across settings, but the availability and accuracy of pocket/interface fields vary with the protocol. Strong-prior results should therefore be interpreted as interface-informed docking results, not as blind docking results.

## B.3. Input Feature Construction

The model represents a protein–ligand complex at two resolutions. At the atom level, it keeps explicit atom states and atom-pair states. At the token level, standard protein residues are represented as residue tokens, while ligand atoms or non-standard chemical components are represented at finer token resolution. An atom-to-token assignment connects the two spaces. Mask and padding variables are omitted from Table 3.

Several composite features deserve explicit definition. The atom feature vector is

$$\mathbf{f}_{\text{atom}} = \big[\mathbf{r}^{\text{ref}}, q_{\text{formal}}, \text{onehot}(\text{element}), 𝟙_{\text{aromatic}}, \text{onehot}(\text{degree}),$$
$$\text{onehot}(\text{hybridization}), \text{onehot}(\textbf{implicit valence}), \text{onehot}(\text{chirality}), 𝟙_{\text{ring3}}, \ldots, 𝟙_{\text{ring8}}\big]. \tag{99}$$

The relative token pair feature for chemical components is

$$\mathbf{f}_{\text{ref}}(i, j) = \big[\text{onehot}(d_{\text{token}}(i, j)), \text{onehot}(\textbf{bond type}_{ij}), 𝟙_{\text{bond}_{ij}},$$
$$𝟙_{\text{double}_{ij}}, 𝟙_{\text{ringbond}_{ij}}, 𝟙_{\text{conjugated}_{ij}}, 𝟙_{\text{aromatic}_{ij}}\big]. \tag{100}$$

The relative-position feature is

$$\mathbf{f}_{\text{rel}}(i, j) = \big[\text{onehot}(\text{clip}(r_i - r_j + r_{\text{max}})), 𝟙_{\text{same chain}}, 𝟙_{\text{same entity}}\big], \tag{101}$$

where $r_{\text{max}}$ is the clipping radius for residue-index offsets, and a separate bin is used for different-chain pairs.

The template-like pair feature is not an external homologous template. It is a structural prior derived from the input/reference protein coordinates:

$$\mathbf{f}_{\text{templ}}(i, j) = \big[\text{dgram}(\mathbf{x}_i^{\text{tok}}, \mathbf{x}_j^{\text{tok}}) \cdot 𝟙_{i,j \in \text{protein}}, \ 𝟙_{i,j \in \text{protein}}\big]. \tag{102}$$

The pocket feature is

$$\mathbf{f}_{\text{pocket},i} = \big[\text{hist}(|\mathbf{x}_i^{\text{tok}} - \mathbf{c}_{\text{pocket}}|) \cdot 𝟙_{i \in \text{protein}}, \ 𝟙_{i \in \text{protein}}, \ 𝟙_{\text{has pocket center}}\big]. \tag{103}$$

The center $\mathbf{c}_{\text{pocket}}$ is constructed according to the evaluation protocol: it is absent in blind settings, perturbed in noisy-pocket settings, exact in exact-pocket re-docking settings, and accompanied by residue-level interface indicators in the strong-prior setting. The dgram and hist setting follow AlphaFold3 (Abramson et al., 2024b).

## B.4. Representation Spaces

The model maintains four latent spaces:

$$\mathbf{A} \in \mathbb{R}^{N_a \times d_a}, \qquad \mathbf{P}_a \in \mathbb{R}^{N_a \times N_a \times d_{aP}}, \qquad \mathbf{S} \in \mathbb{R}^{N_t \times d_s}, \qquad \mathbf{P} \in \mathbb{R}^{N_t \times N_t \times d_z}, \tag{104}$$

where $N_a$ is the number of atoms, $N_t$ is the number of tokens, $\mathbf{A}$ is the atom representation, $\mathbf{P}_a$ is the atom-pair representation, $\mathbf{S}$ is the token/single representation, and $\mathbf{P}$ is the token-pair representation. Atom-level states preserve local chemistry and coordinate-related structure, while token-level states support global protein–ligand communication and long-range geometric reasoning.

## B.5. Atom and Token Embedding

The atom embedder initializes atom states from chemistry and local component geometry:

$$(\mathbf{A}, \mathbf{P}_a) = \text{AtomEmbedder}(\mathbf{f}_{\text{atom}}, \mathbf{r}^{\text{ref}}, \text{atom relations}). \tag{105}$$

Atom-to-token pooling constructs initial token states:

$$\mathbf{S}_i = \text{Pool}\left(\{\mathbf{A}_a : \pi(a) = i\}\right) + \text{Embed}(\mathbf{f}_{\text{type},i}). \tag{106}$$

Initial token-pair states combine relative residue position, ligand/component connectivity, pocket context, and template-like protein geometry:

$$\mathbf{P}_{ij} = F_z\left(\mathbf{f}_{\text{rel}}(i,j), \mathbf{f}_{\text{ref}}(i,j), \mathbf{f}_{\text{pocket},i}, \mathbf{f}_{\text{pocket},j}, \mathbf{f}_{\text{templ}}(i,j)\right). \tag{107}$$

## B.6. MSA-Free Pairformer Representation Trunk

The representation trunk is an MSA-free Pairformer. Each block alternates token attention, token transition, single-to-pair lifting, triangle-based pair refinement, and pair transition. Token attention uses pair bias:

$$\mathbf{S} \leftarrow \mathbf{S} + \text{AttnWithBias}(\mathbf{S}; \mathbf{P}), \tag{108}$$

followed by a token transition:

$$\mathbf{S} \leftarrow \mathbf{S} + \text{Transition}(\mathbf{S}). \tag{109}$$

The token states are lifted back into pair space through an outer-product-mean style operation:

$$\mathbf{P} \leftarrow \mathbf{P} + \text{OPM}(\mathbf{S}). \tag{110}$$

The pair representation is then refined by triangle multiplication and triangle attention in standard co-folding models:

$$
\begin{aligned}
\mathbf{P} &\leftarrow \mathbf{P} + \text{TriMul}_{\text{out}}(\mathbf{P}) + \text{TriMul}_{\text{in}}(\mathbf{P}), \\
\mathbf{P} &\leftarrow \mathbf{P} + \text{TriAttn}_{\text{start}}(\mathbf{P}) + \text{TriAttn}_{\text{end}}(\mathbf{P}), \\
\mathbf{P} &\leftarrow \mathbf{P} + \text{Transition}(\mathbf{P}).
\end{aligned} \tag{111}
$$

Overall, one Pairformer block is

$$
\begin{aligned}
\mathbf{S} &\leftarrow \mathbf{S} + \text{AttnWithBias}(\mathbf{S}; \mathbf{P}), \\
\mathbf{S} &\leftarrow \mathbf{S} + \text{Transition}(\mathbf{S}), \\
\mathbf{P} &\leftarrow \mathbf{P} + \text{OPM}(\mathbf{S}), \\
\mathbf{P} &\leftarrow \mathbf{P} + \text{TriMul}_{\text{out}}(\mathbf{P}) + \text{TriMul}_{\text{in}}(\mathbf{P}), \\
\mathbf{P} &\leftarrow \mathbf{P} + \text{TriAttn}_{\text{start}}(\mathbf{P}) + \text{TriAttn}_{\text{end}}(\mathbf{P}), \\
\mathbf{P} &\leftarrow \mathbf{P} + \text{Transition}(\mathbf{P}).
\end{aligned} \tag{112}
$$

After Pairformer refinement, token and token-pair context are injected back into atom spaces:

$$\mathbf{A} \leftarrow \mathbf{A} + \text{Unpool}\left(F_a(\mathbf{S})\right), \tag{113}$$

and

$$\mathbf{P}_a \leftarrow \mathbf{P}_a + \text{Unpool2D}\left(F_{ap}(\mathbf{P})\right). \tag{114}$$

This feedback step ensures that the atom-level coordinate generator receives both local chemistry and token-level protein–ligand context.

## B.7. Dual-Scale DiT Coordinate Generator

The coordinate generator maps noisy atom coordinates to the raw generative target. It receives conditioning states

$$\mathbf{C} = \{\mathbf{A}, \mathbf{P}_a, \mathbf{S}, \mathbf{P}\}. \tag{115}$$

We distinguish three quantities:

$$\lambda: \text{ Root NSR coordinate}, \qquad s: \text{ native interpolation time}, \qquad \tau_{\text{net}}: \text{ network time input}.$$

The native time $s$ depends on the parameterization, while $\tau_{\text{net}}$ is the scalar variable encoded by the network time encoder. For EDM, $s = \lambda$. For Linear interpolation, $s = u = \lambda/(1 + \lambda)$.

Given noisy coordinates $\mathbf{z}$ and conditioning states $\mathbf{C}$, the DiT predicts a raw target:

$$\mathbf{d}_\theta = \text{DualScaleDiT}(\mathbf{z}, \tau_{\text{net}}; \mathbf{C}). \tag{116}$$

The raw target is decoded into a clean endpoint by a parameterization-specific recovery map:

$$\hat{\mathbf{x}}_0 = g_\theta(\mathbf{z}, \lambda; \mathbf{C}) = \text{Recover}(\mathbf{d}_\theta, \mathbf{z}, \lambda). \tag{117}$$

For Linear interpolation, recovery is

$$\hat{\mathbf{x}}_0 = \mathbf{z}_u - u\mathbf{d}_\theta, \qquad u = \frac{\lambda}{1 + \lambda}. \tag{118}$$

## B.8. Training Objectives

Training objectives are defined in endpoint space. For both EDM/VE-style denoising and Linear interpolation, the raw network output is first decoded into a clean-structure estimate:

$$\hat{\mathbf{x}}_0 = g_\theta(\mathbf{z}, \lambda; \mathbf{C}) = \text{Recover}(\mathbf{d}_\theta, \mathbf{z}, \lambda). \tag{119}$$

**Reconstruction weight.** We use the EDM-style reconstruction-equivalent endpoint weight

$$w_x(\lambda) = \frac{1 + \lambda^2}{\lambda^2}. \tag{120}$$

This weight is inversely proportional to the noise variance, so it naturally emphasizes low-noise structural refinement. If the objective were rewritten directly in raw-target space, the induced raw-target weights would differ by parameterization (§A); by decoding to $\hat{\mathbf{x}}_0$ first, the same $w_x(\lambda)$ applies uniformly.

**Primary reconstruction losses.** Let $d_{\text{align}}(\cdot, \cdot)$ denote a rigid-aligned squared distance (Kabsch alignment over the generated atom set). The all-atom reconstruction loss is

$$\mathcal{L}_x = w_x(\lambda)\, d_{\text{align}}(\hat{\mathbf{x}}_0, \mathbf{x}_0). \tag{121}$$

A ligand-focused aligned reconstruction loss supplements the all-atom objective:

$$\mathcal{L}_{\text{lig}} = w_x(\lambda)\, d_{\text{align,lig}}(\hat{\mathbf{x}}_0, \mathbf{x}_0), \tag{122}$$

where the alignment and MSE are computed over ligand atoms only. $\mathcal{L}_{\text{lig}}$ emphasizes ligand placement while the all-atom $\mathcal{L}_x$ retains global structural context.

**Auxiliary local geometry losses.** In addition to the primary reconstruction losses, we apply two auxiliary local geometry losses that provide fine-grained structural supervision:

$$\mathcal{L}_{\text{sLDDT}} \quad \text{(smooth local distance difference deviation)}, \tag{123}$$

$$\mathcal{L}_{\text{within}} \quad \text{(within-component coordinate consistency)}. \tag{124}$$

$\mathcal{L}_{\text{sLDDT}}$ computes smooth deviations of local inter-atomic distances (within an 8 Å radius cutoff) relative to the ground truth, acting as a differentiable proxy for the smooth lDDT metric (Abramson et al., 2024a). $\mathcal{L}_{\text{within}}$ focuses on preserving local component geometry (bond lengths and angles within each residue or ligand fragment). When coordinates are internally normalized by a data-dependent scale, both auxiliary losses are evaluated after rescaling to physical coordinate units (Ångströms).

**Diffusion pre-training objective.** The combined pre-training objective balances primary and auxiliary terms:

$$\mathcal{L}_{\text{pre}} = \underbrace{4\mathcal{L}_x}_{\text{primary}} + \underbrace{4\mathcal{L}_{\text{sLDDT}} + \mathcal{L}_{\text{within}} + \mathcal{L}_{\text{lig}}}_{\text{auxiliary}}. \tag{125}$$

The coefficient 4 on $\mathcal{L}_x$ and $\mathcal{L}_{\text{sLDDT}}$ reflects their relative scale and ensures that both coarse global pose and fine-grained local geometry contribute meaningfully to the gradient. This objective is shared by EDM/VE-style denoising and Linear interpolation after parameterization-specific endpoint recovery.

**PCR consistency tuning.** After pre-training, we apply Progressive Consistency Regularization (PCR) as a tuning stage to enable reliable few-step inference. PCR samples two adjacent root NSR levels $\lambda > \lambda'$ on the same interpolation path. The corresponding native times are:

$$t = \lambda, \ t' = \lambda' \quad \text{(EDM)}, \qquad u = \frac{\lambda}{1 + \lambda}, \ u' = \frac{\lambda'}{1 + \lambda'} \quad \text{(Linear)}. \tag{126}$$

Let $\mathcal{G}$ denote the set of generated atoms (atoms for which coordinates are predicted, excluding padding and reference-only atoms), with $N_{\text{gen}} = |\mathcal{G}|$. The clean endpoint predictors are

$$\hat{\mathbf{x}}_{0,\lambda} = g_\theta(\mathbf{z}_\lambda, \lambda; \mathbf{C}), \qquad \hat{\mathbf{x}}_{0,\lambda'} = g_{\theta^-}(\mathbf{z}_{\lambda'}, \lambda'; \mathbf{C}), \tag{127}$$

where $\theta^-$ is the EMA of the online parameters $\theta$ and gradients are stopped through the lower-noise branch. The EMA parameters are the same as during inference; no additional teacher rollout is required.

The consistency term quantifies agreement between predictions at two noise levels, expressed as per-atom MSE over $\mathcal{G}$:

$$\mathcal{L}_{\text{cons}} = \frac{1}{N_{\text{gen}}} \sum_{i \in \mathcal{G}} \left\| \hat{\mathbf{x}}_{0,\lambda}^{(i)} - \text{sg}\big[\hat{\mathbf{x}}_{0,\lambda'}^{(i)}\big] \right\|_2^2, \tag{128}$$

with an optional ligand-focused variant $\mathcal{L}_{\text{cons,lig}}$ computed over the ligand atom subset $\mathcal{G}_{\text{lig}}$.

The consistency loss is weighted by the EDM-style inverse-gap weight:

$$w_{\text{cons}}(\lambda, \lambda') = \frac{1}{(\lambda - \lambda')^2}, \tag{129}$$

so that tighter agreement is enforced when adjacent levels are close. For the Linear parameterization, this becomes $w_{\text{cons}}^{\text{lin}}(u, v) = (1 - u)^2 (1 - v)^2 / (u - v)^2$.

All primary and auxiliary reconstruction losses remain active at both adjacent levels during PCR tuning:

$$\mathcal{L}_x = \frac{1}{2} \Big[ \mathcal{L}_x(\hat{\mathbf{x}}_{0,\lambda}, \mathbf{x}_0) + \mathcal{L}_x(\hat{\mathbf{x}}_{0,\lambda'}, \mathbf{x}_0) \Big], \tag{130}$$

and analogously for $\mathcal{L}_{\text{lig}}$, $\mathcal{L}_{\text{sLDDT}}$, and $\mathcal{L}_{\text{within}}$.

The full PCR objective combines the reconstruction-anchored terms with consistency:

$$\mathcal{L}_{\text{PCR}} = \underbrace{4\mathcal{L}_x + 4\mathcal{L}_{\text{sLDDT}} + \mathcal{L}_{\text{within}} + \mathcal{L}_{\text{lig}}}_{\text{primary and auxiliary (both levels)}}$$
$$+ \eta \Big( w_{\text{cons}}(\lambda, \lambda') \, \mathcal{L}_{\text{cons}} + w_{\text{cons}}(\lambda, \lambda') \, \mathcal{L}_{\text{cons,lig}} \Big), \tag{131}$$

where $\eta$ controls the consistency strength. Thus, PCR compresses the trajectory through cross-time consistency while the active reconstruction terms keep the model anchored to the ground-truth molecular structure at both noise levels.

### B.9. Residual-Safe Quantization Implementation

Residual-Safe Quantization is implemented as post-training quantization. Only selected compute-heavy linear transformations inside the Pairformer and token-level DiT blocks are quantized. Residual streams, residual additions, attention softmax,

normalization layers, triangle operations, input embedders, and final structure heads remain in BF16. This preserves the geometry-sensitive pathway while reducing the cost of matrix multiplications.

For W8A16, selected linear weights are quantized to 8-bit integers and activations remain in BF16. For W8A8, both weights and activations are quantized inside selected linear operations. Weight quantization uses per-channel scales. Activation scales are estimated from a calibration set before deployment using the default settings of the quantization backend. After each quantized linear operation, the output is dequantized back to BF16 before residual accumulation. No quantization-aware retraining is used.

### B.10. Overall Model Summary

The complete computation is

$$
\begin{aligned}
(\mathbf{A}, \mathbf{P}_a) &= \text{AtomEmbedder}(\mathbf{f}_{\text{atom}}, \mathbf{r}^{\text{ref}}, \text{atom relations}), \\
(\mathbf{S}_0, \mathbf{P}_0) &= \text{TokenEmbedder}(\mathbf{A}, \mathbf{f}_{\text{type}}, \mathbf{f}_{\text{rel}}, \mathbf{f}_{\text{ref}}, \mathbf{f}_{\text{pocket}}, \mathbf{f}_{\text{templ}}), \\
(\mathbf{S}, \mathbf{P}) &= \text{Pairformer}(\mathbf{S}_0, \mathbf{P}_0), \\
\mathbf{C} &= \{\mathbf{A}, \mathbf{P}_a, \mathbf{S}, \mathbf{P}\}, \\
\mathbf{d}_\theta &= \text{DualScaleDiT}(\mathbf{z}, \tau_{\text{net}}; \mathbf{C}), \\
\hat{\mathbf{x}}_0 &= \text{Recover}(\mathbf{d}_\theta, \mathbf{z}, \lambda).
\end{aligned}
\tag{132}
$$

The model follows the co-folding principle of PhysDock and AlphaFold-style architectures, but removes the explicit MSA-processing track. It retains the components most relevant to docking: atom-level chemistry, token-level global reasoning, pairwise protein–ligand communication, OPM-style single-to-pair lifting, triangle-based geometric consistency, dual-scale coordinate refinement, and reconstruction-anchored few-step generative training.

## C. Architecture Configuration, Training, and Inference Details

This section summarizes the concrete model configuration, training protocol, and inference setup used in our main experiments. The goal of this configuration is not to maximize model size, but to preserve the core co-folding operations required for docking while keeping the system suitable for low-latency deployment.

**Architecture configuration.**  We instantiate the PhysDock (Zhang et al., 2025) framework with a compact MSA-free co-folding configuration. The channel dimensions of the Single, Pair, Atom, and Atom-Pair representations are set to

$$
d_s = 512, \qquad d_z = 128, \qquad d_a = 128, \qquad d_{ap} = 16. \tag{133}
$$

The representation trunk contains 8 MSA-free Pairformer blocks. Each block retains the operations most relevant to docking: token-level attention with pair bias, OPM-style single-to-pair lifting, triangle-based pair refinement, and pair transition. The coordinate-generation trunk uses a lightweight 6-block token-level DiT stack for dual-scale atom–token–atom coordinate refinement.

This configuration is deliberately asymmetric. The Pairformer trunk is kept expressive enough to build protein–ligand context and interface geometry, while the DiT trunk is kept lightweight because it is repeatedly evaluated during coordinate generation. This reflects the accuracy–latency principle used throughout the paper: preserve the computation that determines structural fidelity, and shorten the computation that limits deployment.

**Training setup.**  Training is conducted on 8 NVIDIA H20 GPUs in FP32, with TF32 acceleration enabled for matrix multiplications. Due to the memory cost of Pairformer-based pair representations, the physical mini-batch size is set to 1 complex per GPU. To stabilize diffusion training under this memory constraint, we use a *diffusion batch* strategy: for each physical complex, we generate 48 independently noised and randomly rigid-transformed views. This increases the number of diffusion supervision signals per optimization step without increasing the number of distinct complexes loaded into GPU memory.

Input complexes are spatially cropped to at most 256 tokens. The diffusion pre-training stage is run for 200k global steps, taking approximately 60 hours on the 8-GPU H20 cluster. We use Adam with

$$
\beta_1 = 0.9, \qquad \beta_2 = 0.95, \tag{134}
$$

and a peak learning rate of $1 \times 10^{-3}$. The learning rate is linearly warmed up for the first 1,000 steps and then decayed exponentially by a factor of $\gamma = 0.99$ every 5,000 steps. For PCR tuning, we use a 100k-step tuning budget on top of the diffusion-pretrained checkpoint.

**PCR tuning protocol.** PCR tuning uses the same data split and structural conditioning protocol as diffusion pre-training. For each training sample, two adjacent root NSR levels are selected from the progressive grid described in Section 4.2. The higher-noise prediction is trained to remain consistent with the stop-gradient lower-noise prediction, while reconstruction and local geometry losses remain active. This keeps the few-step predictor anchored to the molecular structure rather than turning consistency learning into a purely self-distillation objective.

**Inference setup.** For all inference experiments, we use exponential moving average (EMA) weights with decay rate 0.999. EMA stabilizes the final predictor and reduces sensitivity to optimization noise, which is especially important for few-step generative sampling. Unless otherwise specified, inference follows the NSR-based sampling schedule, with the parameterization-specific recovery map used to decode the raw network output into a clean structural prediction.

Accuracy and latency are evaluated under different but explicitly reported protocols. Accuracy experiments use the sample-and-rank setting described in Section 5.1. Latency experiments use the deployment-oriented setting in Section 5.5: generating 5 conformations for a representative 256-token complex under the same input-shape and output-count protocol. This separates pose-quality evaluation from pure runtime measurement while keeping the deployment benchmark reproducible.

**Quantized deployment.** For the low-latency deployment setting, we apply Residual-Safe Quantization after training. Only selected compute-heavy linear transformations inside Pairformer and token-level DiT blocks are quantized. Residual streams, residual additions, attention softmax, normalization layers, triangle operations, input embedders, and final structure heads remain in BF16. For W8A16, selected weights are quantized to 8-bit integers and activations remain in BF16. For W8A8, both selected weights and activations are quantized inside linear operations, with per-channel weight scales and calibration-based activation scales provided by the quantization backend. Quantized outputs are dequantized back to BF16 before residual accumulation.

This deployment configuration preserves the geometry-sensitive pathway while accelerating the matrix-multiplication-heavy branches. Together with single-pass representation recycling and low-NFE PCR sampling, it defines the sub-second operating point used in the main latency benchmark.

# D. Exploratory Experiments

This appendix reports exploratory ablations that guided the final design. These experiments were not intended to introduce additional model components; rather, they clarify which factors materially affect docking accuracy and which engineering choices can be simplified without loss of performance. The main conclusion is consistent across studies: in co-folding-based docking, the dominant determinants of accuracy and latency are the information level of the structural prior, the stability of few-step generative dynamics, and the cost of repeated inference blocks.

## D.1. Optimization Landscape

We first investigated whether training was bottlenecked by optimizer choice or by a difficult optimization landscape. We compared standard Adam, AdamW, and the recently proposed momentum-orthogonalized optimizer Muon (Liu et al., 2025; Jordan et al.). Muon consistently produced lower gradient norms than AdamW, suggesting a smoother optimization trajectory. However, the convergence curves and final validation metrics were nearly unchanged across optimizers.

This result indicates that, for our co-folding docking setting, optimizer choice is not the primary bottleneck once the representation trunk, coordinate generator, and reconstruction losses are fixed. Unlike some large-scale language modeling regimes where optimizer design can substantially affect scaling behavior, our docking model appears to be more strongly governed by input structural information and generative parameterization. We therefore use Adam as the default optimizer in the final system and focus subsequent design effort on trajectory compression, structural conditioning, and deployment efficiency.

**D.2. Information-Conditioning Hierarchy**

A central empirical observation is that docking accuracy depends strongly on the amount of structural information provided at inference time. This does not imply that strong-prior protocols are invalid; rather, it means they answer a different scientific question from blind docking. Blind docking evaluates global search and pocket localization. Pocket-centered docking evaluates pose generation under a specified binding region. Interface-informed docking evaluates high-fidelity pose refinement when residue-level interaction context is available. These regimes should not be collapsed into a single undifferentiated benchmark.

To quantify this effect, we evaluate the same model under a four-level information-conditioning hierarchy. The levels range from blind global search to strongly interface-informed re-docking:

- **Level 1: Blind Docking.** The model receives no spatial prior about the binding site. This is the hardest setting and corresponds most closely to de novo blind docking.

- **Level 2: Noisy Pocket Center.** The model receives an approximate pocket center, constructed by applying a random shift within a 6 Å radius to the reference pocket center. This approximates practical settings where a binding region is estimated from homologues, pocket predictors, weak experimental evidence, or prior biological knowledge.

- **Level 3: Exact Pocket Center.** The model receives the exact reference pocket center. This corresponds to a strong re-docking prior and is common in pocket-specified benchmark settings.

- **Level 4: Interface-Informed Strong Prior.** In addition to the pocket center, the model receives residue-level interface indicators, such as residues within a contact radius of the ligand. This setting is substantially more constrained and should be compared with surface- or interface-informed methods rather than blind docking methods.

We trained models with the EDM parameterization across these four regimes and evaluated docking success using a 20-step inference trajectory. Table 4 shows a clear monotonic trend: success rate increases with the amount of structural information. This validates the information-level-matched evaluation principle used in the main text. Strong-prior results are meaningful for evaluating interface-informed pose refinement, but they should not be interpreted as blind docking performance. Conversely, blind and noisy-pocket results better reflect the model's ability to localize and generate poses under weaker structural guidance.

These results have two implications. First, our strongest accuracy number should be read under the Level-4 interface-informed protocol and compared with methods that use comparable surface or interface information. Second, improving Level-1 and Level-2 performance remains the more realistic route toward prospective blind docking. Thus, the hierarchy does not weaken the main result; it clarifies the operating regime in which each result should be interpreted.

**Bridging the flexibility gap.** Our current semi-flexible docking setting keeps the receptor backbone fixed and focuses on ligand pose and local pocket geometry. This is computationally efficient, but it cannot capture large-scale receptor conformational changes. A practical extension is ensemble docking: running the model over multiple receptor conformations sampled from molecular dynamics, experimental ensembles, or generative structure priors. This expands the conformational search space while preserving the low-latency advantage of the model, offering a pragmatic bridge between rigid efficiency and fully flexible co-folding fidelity.

**D.3. Final Projection Layer of the DiT Decoder**

We next examined the final projection layer of the coordinate generator. We compared two designs: a **standard linear projection**, similar to the output projection used in AlphaFold3-style diffusion modules (Abramson et al., 2024b), and an **adaptive modulation layer**, commonly used in latent DiT architectures (Yao et al., 2025; 2024; Li & He, 2025; Wang et al., 2025b). The adaptive variant injects timestep information into the final projection through affine modulation.

Both designs achieved similar final docking accuracy and convergence speed. However, the adaptive variant occasionally produced late-stage gradient instability, likely because the multiplicative factor $(1 + \gamma(t))$ can amplify small coordinate-level corrections near low noise levels. Since the additional expressivity did not translate into measurable docking gains, we use the simpler standard projection in the final model. This choice is consistent with the broader design principle of the paper: preserve stable structural computation and avoid components that increase numerical sensitivity without improving accuracy.

*Listing 1.* PyTorch-style pseudocode comparing the standard projection and the adaptive modulation layer. The adaptive variant introduces a timestep-dependent multiplicative factor.

```
# 1. Standard linear projection.
# norm can be RMSNorm or LayerNorm.
def forward_standard(x):
    return linear(norm(x))

# 2. Adaptive modulation.
# norm can be RMSNorm or LayerNorm.
def forward_adaptive(x, t):
    gamma, beta = mlp(t).chunk(2)
    x = norm(x) * (1 + gamma) + beta
    return linear(x)
```

### D.4. Explicit Coordinate Modulation

We also explored whether injecting instantaneous coordinate information into the DiT attention mechanism improves spatial awareness. Motivated by relative positional encodings and rotary position embeddings (Su et al., 2021; Zeng et al., 2025), we applied 3D Rotary Positional Embeddings to attention queries and keys using the noisy atomic coordinates $\mathbf{x}_t$.

This explicit coordinate modulation did not improve convergence or docking fidelity. We attribute this to two factors. First, during high-noise stages, $\mathbf{x}_t$ is dominated by corruption, so using it as an attention prior can inject unstable spatial signals. Second, the conditional representation already contains rich geometric information through token-pair features, pocket context, and triangle-updated pair representations. As a result, additional coordinate-dependent attention modulation provides limited marginal information.

This ablation suggests that, in our co-folding architecture, the decoder is not the main source of spatial knowledge. Most structural context is already encoded in the conditional pair and token representations. We therefore omit explicit 3D RoPE from the final model and rely on the Pairformer-conditioned DiT for coordinate refinement.

### D.5. Thermodynamically-Informed Noise Priors

Standard diffusion models typically start from an isotropic Gaussian prior,

$$p(\mathbf{x}_T) \sim \mathcal{N}(\mathbf{0}, \mathbf{I}). \tag{135}$$

Prior work has suggested that structured priors can help generative sampling by aligning the initial distribution with the data manifold, such as adaptive priors in PriorGrad (gil Lee et al., 2022) or harmonic priors in EigenFold (Jing et al., 2023). Motivated by this idea, we tested a thermodynamically-informed Gaussian prior,

$$p(\mathbf{x}_T) \sim \mathcal{N}(\mathbf{0}, \boldsymbol{\Sigma}), \tag{136}$$

where $\boldsymbol{\Sigma}$ encodes pairwise residue correlations derived from Boltzmann-style statistics of the receptor structure. The goal was to break the independence assumption of the initial noise and bias sampling toward physically plausible structural directions.

Empirically, this structured prior did not produce a statistically significant improvement in docking success. This suggests that, in strongly conditional docking, the learned conditional vector field dominates the sampling trajectory after the first few denoising steps. The initial noise geometry matters less than the quality of the conditional protein–ligand representation and the stability of the generative update. We therefore keep the isotropic Gaussian prior in the final model.

This negative result is useful: it indicates that over-engineering the noise prior is less effective than improving the conditional encoder, trajectory parameterization, and low-NFE consistency. It also supports the streamlined design used in the main text, where efficiency is gained by simplifying the generative path rather than adding hand-crafted physical priors.

# E. Limitations

This work focuses on establishing a controlled accuracy–latency frontier for co-folding-based molecular docking. Several limitations remain.

**Benchmark scope and data scaling.**   Our experiments are conducted under a unified evaluation protocol on curated PDBBind2020 complexes. Whenever feasible, we train or re-train baselines under the same data split and evaluation setting to reduce confounding factors. However, we do not claim that this benchmark exhaustively covers all docking regimes. Docking performance can be strongly affected by training-set scale, dataset overlap, pocket-definition choices, and the strength of structural priors available at inference time. For this reason, we emphasize controlled comparison under an explicit information-level protocol rather than pursuing the largest possible benchmark number. Future work will release stronger models trained on larger and more diverse datasets, and will evaluate them across broader docking benchmarks and prospective settings.

**Quantization design space.**   Residual-Safe Quantization substantially reduces memory footprint and improves deployment efficiency, but the current configuration is not necessarily optimal. The choice of weight-only versus weight–activation quantization, static versus dynamic activation scaling, calibration data, and parameter-modulation strategy can all affect both speed and structural fidelity. In particular, aggressive quantization may preserve global ligand placement while perturbing fine-grained local geometry, such as side-chain torsions. A more systematic search over quantization schemes, calibration procedures, and hardware-specific kernels may further improve the accuracy–latency trade-off and enable near-lossless compression of co-folding docking models.

**Scaling with input length.**   Co-folding architectures rely heavily on pair representations and pairwise geometric reasoning. As a result, their inference cost does not grow linearly with the number of input tokens. This limits direct scaling to very long proteins or large biomolecular assemblies. Our deployment benchmark uses 256 tokens, which covers the majority of pocket-centered docking scenarios where the binding region is known or can be cropped. Nevertheless, long-context docking remains an open challenge. Future architectures may need to reduce the dependence on dense pair modules, for example through sparse pair reasoning, adaptive cropping, or pair-light designs inspired by emerging models such as SimpleFold. Such directions could make co-folding-style docking more scalable for long sequences while retaining the geometric fidelity needed for local protein–ligand interfaces.

*Table 3.* **Non-mask features used by the model.** Training-only supervision variables are separated from inference-time conditioning and bookkeeping variables. Pocket and interface fields are instantiated according to the evaluation protocol.

| Group | Feature | Source and composition |
|---|---|---|
| **Training supervision** | $\mathbf{x}_{gt}$ | Ground-truth all-atom coordinates. Used for endpoint reconstruction, aligned MSE, local geometry losses, and validation metrics. |
| | $\mathbf{x}_{gt}^{tok}$ | Token-center coordinates derived from the reference structure. For protein residues this is typically the CA coordinate; for non-protein components it is a component or atom-center coordinate. Used to construct supervised structural pair features and pocket context during data preparation. |
| **Atom chemistry** | $\mathbf{f}_{atom}$ | Concatenation of reference coordinate, formal charge, element one-hot, aromaticity, atom degree one-hot, hybridization one-hot, implicit-valence one-hot, chirality one-hot, and ring-membership indicators for small rings. |
| | $\mathbf{r}^{ref}$ | Reference atom coordinates from the component dictionary. These initialize local component geometry. |
| | $r_{vdW}$ | Element-dependent van der Waals radius. Used for steric analysis and geometry-aware validation. |
| **Token identity** | residue/component type | One-hot identity over standard residue/component vocabulary. Protein residues, ligand atoms, and non-standard components are mapped into the token vocabulary. |
| | protein/ligand indicators | Binary indicators specifying whether each token or atom belongs to receptor protein or ligand. |
| | chain/entity metadata | Residue index, chain/asymmetry identifier, and entity identifier. Used for relative-position and chain/entity pair features. |
| **Atom–token organization** | atom-to-token map $\pi(a)$ | Assigns atom $a$ to token $\pi(a)$. Used for pooling atom features into tokens and unpooling token features back to atoms. |
| | atom-to-residue map | Assigns each atom to its source residue or chemical component. Used for atom relation features and local component structure. |
| | split sizes | Number of atoms assigned to each token. Used by pooling and unpooling operators. |
| **Ligand/component connectivity** | bond matrix | Atom-level bonded relations within ligand or non-standard chemical components. |
| | relative token pair features | Component-level pair features: token distance, bond type, bond existence, double-bond flag, ring-bond flag, conjugation flag, and aromatic-bond flag. |
| **Protein and pair geometry** | relative residue-position features | Clipped residue-index offset for same-chain pairs, a special bin for different-chain pairs, plus chain-same and entity-same indicators. |
| | template-like pair features | Pairwise structural features derived from input/reference protein coordinates: distance-bin features between token centers, multiplied by protein–protein indicators, plus a protein-pair indicator. |
| | atom relation metadata | Chain-same, residue-same, and atom-same indicators used to define local atom-pair relations. |
| **Pocket/interface context** | pocket feature vector | Distance-to-pocket histogram for protein tokens, protein/ligand type indicator, and an indicator specifying whether a pocket center is available. |
| | pocket indicator | Indicates residues within the selected pocket radius or from a provided/estimated pocket specification. |
| | interface indicator | Optional residue-level interface cue used only in the interface-informed strong-prior protocol. |
| **Bookkeeping** | component names, residue indices, atom indices | Used to map model outputs back to molecular components and write predicted structures. Not used as learned continuous features. |

*Table 4.* **Effect of Input Information Level on Docking Success.** We evaluate docking success rate, defined by ligand RMSD $< 2.0$ Å, across four information-conditioning regimes on 428 test cases. The monotonic improvement shows that prior strength is a major determinant of docking difficulty, motivating information-level-matched comparisons.

| Information Level | Prior Description | Success Count | Success Rate |
|---|---|---|---|
| Level 1: Blind Docking | Zero spatial prior | 225 / 428 | 52.6% |
| Level 2: Noisy Pocket | Center $\pm$ 6 Å | 253 / 428 | 59.1% |
| Level 3: Exact Center | Exact pocket center | 316 / 428 | 73.8% |
| Level 4: Interface-Informed | Center + residue-level interface cue | 353 / 428 | **82.48%** |

