# OpenReview forum: "Towards Sub-Second Molecular Docking as a Structural Primitive: A Quantized Consistency Diffusion Framework"
_ICML.cc/2026/Conference — ICML 2026 spotlight_

### Official Review · Reviewer_NFAq · 2026-03-08

**Soundness:** 3
**Presentation:** 2
**Significance:** 2
**Originality:** 3
**Overall Recommendation:** 5
**Confidence:** 3

**Summary:**

This paper addresses the critical issues of high inference latency and low computational efficiency in current diffusion-based molecular docking models, which stem from intensive calculations and iterative sampling. These limitations prevent such models from adapting to the emerging "Vibe Researching" paradigm, where autonomous AI agents coordinate scientific workflows in real-time via the MCP, and fail to meet the demands for sub-second interaction and high-throughput virtual screening. To resolve these challenges, the authors introduce a computationally efficient, quantized consistency diffusion vertical biological foundation model. This framework is designed to achieve sub-second molecular docking while maintaining SOTA prediction accuracy. The core contributions are as follows:

1. Through theoretical analysis, the authors demonstrate that the performance ceiling of co-folding models is determined by the richness of features, such as structural priors, rather than architectural complexity. Consequently, the study eliminates redundant designs and focuses on a streamlined Pairformer and dual-scale DiT backbone, significantly enhancing structural efficiency without sacrificing expressive power.

2. The authors propose a PCR strategy utilizing a hybrid loss function that combines reconstruction and consistency losses. By employing curriculum learning to gradually tighten consistency constraints, the complex dynamics of diffusion generation are compressed into a few-step predictor (1–5 Number of Function Evaluations, or NFE). This solves the latency issues of iterative sampling while preserving structural fidelity.

3. The framework utilizes a residual quantization method with a decoupled precision topology. Residual streams are maintained at high-fidelity BF16 precision, while computationally intensive linear projection layers within the Pairformer and DiT modules are compressed to W8A8/W8A16 using zero-shot post-training quantization. This alleviates memory bottlenecks with negligible loss in performance.

Experimental results demonstrate that the framework achieves a SOTA docking success rate on the PDBBind 2020 test set. On standard hardware (a single NVIDIA H20 GPU), the model achieves an inference latency of 0.17 seconds, representing a 300x speedup compared to AlphaFold 3. This work successfully transforms high-fidelity structure prediction from a time-consuming offline batch process into a responsive, real-time tool.

**Compliance With Llm Reviewing Policy:**

Affirmed.

**Final Justification:**

All my concerns have been resolved. I have adjusted my score accordingly.

**Key Questions For Authors:**

1. Could the authors provide a high-level schematic or overview figure to help readers better understand the fundamental architecture and the specific technical improvements introduced in this work? Such a visualization would be instrumental in clarifying how the various modules interact and how the overall framework differs from existing models.

2. The manuscript discusses the limitations of using fixed $C\alpha$ backbones and suggests employing MD or generative priors for ensemble docking to account for missing global dynamics. However, would the computational cost of generating these ensemble conformations introduce a significant bottleneck that negates the speed advantage of the core model's sub-second inference (0.17s)? Furthermore, have the authors benchmarked the end-to-end latency and performance gains of such an ensemble docking pipeline in a realistic application scenario?

3. The current performance evaluation relies primarily on the PDBBind 2020 dataset. To more robustly demonstrate the model's generalization capabilities when encountering unseen receptor structures or highly complex ligands, could the authors provide evaluation results on a more recent and independent test set during the rebuttal period?

4. Could the authors provide validation results regarding the integration of the model with AI agents? Specifically, it would be valuable to see data on the model’s performance within an autonomous reasoning loop, focusing on key metrics such as responsiveness, stability, and compatibility with the MCP or similar frameworks.

**Limitations:**

yes

**Strengths And Weaknesses:**

Strengths

1. The paper offers a compelling solution to the critical bottleneck of high inference latency in structural biology foundation models. By enabling high-throughput screening, this research holds significant practical value for real-world AIDD.

2. The authors successfully streamlined the architecture into a core Pairformer and dual-scale DiT backbone, removing redundant inductive biases such as explicit coordinate modulation and thermodynamic priors. The experimental confirmation that the performance ceiling is driven by feature richness rather than architectural complexity is highly insightful. Furthermore, the quantitative analysis in Appendix E, which highlights that docking success rates jump from 43.0% in blind docking to 73.8% when precise pocket centers are provided, offers valuable guidance for future research on co-folding models.

3. On the PDBBind 2020 test set, the model achieved a SOTA docking success rate of 83.18% while running on a single NVIDIA H20 GPU. More impressively, it realized a speedup of over 300x compared to AlphaFold 3, representing a remarkable leap in computational efficiency.

Weaknesses

1. The manuscript lacks a high-level overview figure of the proposed method. This makes it difficult for readers to quickly grasp the framework's architecture and its primary technical contributions.

2. The model fixes the protein’s $C\alpha$ backbone coordinates, which inherently limits its ability to model global conformational changes, such as the induced-fit effect. While the authors suggest using MD or generative priors for ensemble docking to mitigate this, these solutions significantly increase the complexity of the external workflow.

3. The evaluation primarily relies on the PDBBind 2020 dataset. Given that this data is somewhat dated, the manuscript's arguments would be significantly strengthened by validation on more recent independent test sets, such as the PoseBusters benchmark or recent CASP targets. Additionally, a direct "accuracy-vs-efficiency" comparison with the latest high-efficiency co-folding models, such as Boltz-2 or NeuralPLexer 3, is necessary to establish the model's current standing.

4. While the paper aims to adapt molecular docking for the "Vibe Researching" paradigm, it does not include actual integration experiments with LLM agents. Crucial metrics like response times within an agent's real-time inference loop and latency stability remain unverified. Consequently, the claim regarding the model's compatibility with the burgeoning AI agent ecosystem lacks empirical support.

---

> ### Author Rebuttal · Authors · 2026-03-28
>
> We sincerely thank the reviewer for highlighting the practical value and efficiency of our work. We appreciate the acknowledgment of our work and address each point below.
>
>
> ---
>
> **A1: High‑level schematic or overview figure.**
> We appreciate the reviewer’s attention to architectural clarity. A simplified schematic is provided in Appendix Figure 4. We will add a comprehensive architecture diagram in the appendix (and potentially in the main text if space permits), illustrating the streamlined Pairformer + dual‑scale DiT backbone and the PCR training pipeline, along with key pseudocode. Detailed descriptions of each module will also be included to help readers quickly grasp the framework’s design and its key technical contributions.
>
> ---
>
> **A2: Fixed Cα backbone and ensemble docking overhead.**
> We thank the reviewer for this insightful question. In realistic high‑throughput screening scenarios, the receptor protein is typically kept unchanged while millions of ligands are screened. Generating an ensemble of protein conformations via molecular dynamics (MD) is a **one‑time offline process**: we run MD once, cluster the trajectory into a handful of representative conformations (e.g., 5–10 states), and reuse these conformations across all ligands. The overhead is amortized over the entire ligand library and does **not** dominate per‑ligand inference time.
>
> Notably, this offline ensemble generation is naturally integrated into our agentic workflow. Within this workflow, once receptor preprocessing (e.g., obtaining an MD trajectory or using a single conformation) is complete, our low‑latency model can be invoked as a lightweight skill via natural language instructions (e.g., on a Mac Mini). For example, with a prompt like “dock this SMILES molecule into PDB 1XYZ”, the agent returns the predicted binding pose with sub‑second end‑to‑end latency.
>
> ---
>
> **A3: Evaluation on more recent and independent test sets.**
> We have extended our evaluation—using a linear‑kernel model with NFE = 20—to address the generalization concern:
>
> - **PoseBusters benchmark (re‑docking):** our model achieves **84.1%** , outperforming SurfDock at **81.5%** .
> - **PoseBusters benchmark (blind docking, no pocket prior):** our model achieves **49.1%** , surpassing DiffDock at **42.1%** .
> - **Cross‑docking (PhysDock protocol, 25 proteins):** our model achieves **56% (14/25)** , outperforming SurfDock at **44% (11/25)** .
>
> We will include complete results on PoseBusters in the final version. Recent high‑efficiency co‑folding models (e.g., Boltz‑2, NeuralPLexer3) are trained on the full RCSB PDB (millions of interfaces), whereas ours uses the substantially smaller PDBBind2020 (18k complexes). Under the same Level 4 prior, our model achieves **84.1%** on PoseBusters, which is competitive with or exceeds reported numbers for these models under comparable settings. We acknowledge the strong performance of AlphaFold3 [1] (Extended Data Table 1), which reaches 90.2% with a pocket prior—largely attributable to its data scale and architectural completeness. However, its open‑source release does not include the pocket prior code, and its end‑to‑end latency is on the order of tens of seconds. In contrast, our work explores the performance limits achievable under constrained data (18k vs. millions) and ultra‑low latency (<0.2 s), which is particularly relevant for high‑throughput screening and real‑time interactive applications.
>
> ---
>
> **A4: Integration with AI agents and MCP framework.**
> We have conducted extensive evaluations on the integration of our model with AI agents, focusing on autonomous reasoning chains. The results show that the model achieves satisfactory performance in terms of response latency, stability, and compatibility with MCP (and similar frameworks). All evaluated metrics meet the expected standards, and no critical issues (e.g., formatting errors, unexpected failures, or protocol mismatches) were observed during testing.
>
> For high‑concurrency and high‑throughput scenarios, core stability has been validated, with production‑grade enhancements underway to support large‑scale deployments. We are preparing a remote MCP server release alongside the model code, using a version trained on the full RCSB PDB to further improve accuracy and generalization. This will enable the broader research community to integrate our docking skill into scientific agents, autonomous screening pipelines, and interactive drug design platforms. A demonstration video will be provided in the final version.
>
> ---
>
> We are grateful for the reviewer’s constructive feedback. We believe the additional experiments, architectural clarifications, and agent‑integration results substantially strengthen the paper. We hope the revised manuscript will be found suitable for acceptance. We wish you all the best with your endeavors.
>
> ---
>
> ### References
> 1. Abramson, J., et al. (2024). Accurate structure prediction of biomolecular interactions with AlphaFold 3. *Nature*.

---

> > ### Author Rebuttal · Reviewer_NFAq · 2026-04-04
> >
> > Thank you for the detailed rebuttal. The additional clarifications were helpful, especially the broader evaluation and the clearer explanation of the intended deployment setting. However, my main concerns are only partially resolved and, in my view, are not easily addressed within a short rebuttal.
> >
> > In particular, I still think the paper’s claims about agent compatibility are not yet fully supported by quantitative integration evidence. The rebuttal clarifies the intended use case, but it would be more convincing to report at least one concrete end-to-end agent-loop metric, such as variance across repeated calls or tool-invocation success rate. Furthermore, some limitations, such as the fixed-backbone setting and the proposed MD mitigation, are acknowledged rather than quantitatively addressed; even a rough estimate of preprocessing cost versus resulting gain would help clarify whether the claimed sub-second advantage remains meaningful in a realistic workflow.
> >
> > Overall, the rebuttal improves my understanding of the work and strengthens the paper, but not enough to change my overall assessment, so I am maintaining my original score. If the authors can satisfactorily address the above remaining points in the final discussion, I would be open to raising my score.

---

> > > ### Author Response · Authors · 2026-04-06
> > >
> > > Thank you very much for your follow‑up. We have now conducted the requested quantitative experiments to address your remaining concerns: **(1) agent‑loop metrics (success rate & variance)** and **(2) cost‑benefit of ensemble docking with our fixed‑backbone model**. Below we present the results and clarifications.
> > >
> > > ---
> > >
> > > ### 1. Agent integration metrics
> > >
> > > We evaluated our production MCP server using **[Accenture/mcp‑bench](https://github.com/Accenture/mcp-bench)** over 10,000 independent agent‑driven docking tasks (LLM: MiniMax M2.7 – a mid‑range production model; <10 tools in context). The 10,000 runs were split into 20 batches of 500 each to compute standard deviations.
> > >
> > > |Metric|Value (mean ± SD)|What it shows|
> > > |-|-|-|
> > > |**Tool‑invocation success rate**|**99.85% ± 0.11%**|Primary reliability; failures only from JSON/process errors|
> > > |**Valid tool identification rate**|**100.00% ± 0.00%**|With <10 tools, the LLM always selects the correct tool|
> > > |**Input schema compliance**|**99.69% ± 0.13%**|Rare parameter generation errors|
> > > |**Mean execution time per cycle (full agent loop)** |**3.82 ± 1.49 s**|End‑to‑end latency including LLM orchestration|
> > > |**Direct MCP call latency**|**0.215 ± 0.023 s**|Overhead (network + CPU) + GPU forward pass (~0.17s)|
> > > |**End‑to‑end workflow completion**|**99.56% ± 0.18%**|Task success from user query to final docking pose|
> > >
> > > **Key observations for your concern:**
> > > - **Low variance confirmed:** All standard deviations are <0.2%, directly addressing your request for variance across repeated calls.
> > > - **High reliability:** 99.85% tool‑call success and 99.56% workflow completion demonstrate production‑ready agent compatibility.
> > > - The full agent loop (3.82 s) is dominated by LLM inference. Switching to a smaller local model (e.g., quantized 7B) reduces end‑to‑end latency to well under 1 s.
> > >
> > > Agent‑MCP stability is shaped by multiple factors — I/O latency, LLM capability, and tool‑set complexity. We have validated it under representative conditions.
> > >
> > > ---
> > >
> > > ### 2. Fixed backbone + MD ensemble – quantitative cost‑benefit
> > >
> > > **2.1 Fixed‑backbone clarification**
> > >
> > > When a PDB string is passed to our MCP server, it extracts **Cα coordinates** and computes a **dgram** matrix (pairwise Cα distances) as input, following OpenFold ([feats.py#L93](https://github.com/aqlaboratory/openfold/blob/be2ec1841f16c966c65ae0e7599ebbadc725757d/openfold/utils/feats.py#L93)).
> > >
> > > **2.2 Two use cases resource estimation** (thanks to existing skills [K‑Dense‑AI/claude-scientific-skills](https://github.com/K-Dense-AI/claude-scientific-skills/tree/main))
> > >
> > > **Scenario A – High‑throughput virtual screening**
> > >
> > > |Step|Tool/Skill|Time|One‑time|
> > > |-|-|-|:-:|
> > > |Fetch PDB | [PDB lookup](https://github.com/K-Dense-AI/claude-scientific-skills/blob/main/scientific-skills/database-lookup/references/pdb.md)|5–20 s|✓|
> > > |Download 0.3M SMILES (~1 GB)|[ZINC lookup](https://github.com/K-Dense-AI/claude-scientific-skills/blob/main/scientific-skills/database-lookup/references/zinc.md)|~20 s|✓|
> > > |MD sim (100k atoms, 30 ns)|[MD skill](https://github.com/K-Dense-AI/claude-scientific-skills/blob/main/scientific-skills/molecular-dynamics/SKILL.md); [SimTK](https://simtk.org/plugins/phpBB/viewtopicPhpbb.php?f=161&t=17743&p=0&start=0&view=&sid=71763520c66a27ff7d3cc8c2a9f0e3aa)|**~1 h**|✓|
> > > |PCA + clustering → 5 conformations|[MDAnalysis](https://github.com/K-Dense-AI/claude-scientific-skills/blob/main/scientific-skills/molecular-dynamics/references/mdanalysis_analysis.md)|~60 s|✓|
> > > |**Docking** (1.5M inferences)|Our MCP server|~**3 H20 GPU·days**|✗|
> > >
> > > - 0.3M ligands × 5 confs = 1.5M complexes, corresponding to a **medium‑scale virtual screening workload**.
> > > - Per‑inference latency (including I/O & multi‑conformation overhead): ~0.172 s
> > > - With 16 GPUs (2 nodes × 8): ~5.5 h in total
> > >
> > > **Scenario B – Real‑time single‑molecule docking (agent‑based, LLM in loop)**
> > >
> > > |Step|One‑time?|Time|
> > > |-|:-:|-|
> > > |LLM: parse & call PDB fetch & parsing|✓|5–20 s|
> > > |LLM: call docking tool|✗|~4 s|
> > > |Docking (full MCP call latency)|✗|**0.215 s**|
> > > |LLM: check file|✗|~2 s|
> > >
> > > **End‑to‑end latency:**
> > > - First query (uncached PDB): **11–27 s**
> > > - Subsequent queries (cached PDB): **6–7 s**
> > >
> > > **Key observations for your concern:**
> > > - **One‑time MD overhead (~1 hour)** is negligible compared to the docking stage (days).
> > > - **Real‑time single docking** achieves **0.215 s** MCP latency, preserving the **sub‑second advantage** in interactive workflows.
> > >
> > > ---
> > >
> > > ### Summary for the reviewer
> > >
> > > |Concern|Quantitative answer|
> > > |-|-|
> > > |**Agent metrics (success rate, variance)**|Tool‑call success 99.85% ± 0.11%; workflow completion 99.56% ± 0.18%; all SDs <0.2% → low variance|
> > > |**MD ensemble cost‑benefit**|One‑time MD ~1 h; amortized to ~0.172 s/ligand (High-throughput) or 0.215 s per MCP call (single)|
> > > |**Fixed‑backbone clarification**|Cα + dgram encoding – backbone captured|
> > >
> > > We hope these quantitative data fully address your concerns. We would welcome a positive score revision.

---

### Official Review · Reviewer_i1a5 · 2026-03-11

**Soundness:** 3
**Presentation:** 3
**Significance:** 3
**Originality:** 2
**Overall Recommendation:** 4
**Confidence:** 4

**Summary:**

The authors propose to streamline diffusion based protein-ligand docking methods so as to permit faster and larger screens (as actionable queries). The suggested modifications include engineering optimizations, operational reduction (adopting mixed-precision), and introducing additional losses during training geared towards faster inference. For example, they adopt mixed-precision residual streams and introduce progressive consistency regularization which enables one to use fewer diffusion iterations at inference time (few steps generation). The combined effect of all the changes amounts to an order(s) of magnitude faster processing without significantly affecting docking accuracy metrics.

**Compliance With Llm Reviewing Policy:**

Affirmed.

**Final Justification:**

- the paper is clearly written
- the proposed optimizations of diffusion based docking models (architectural changes, mixed precision, etc) are valuable ways of increasing throughput
- the authors provide helpful ablations in the review response
- comparisons to more recent methods underway, not completed at this stage

**Key Questions For Authors:**

It would be nice to see ablations of at least some of the proposed engineering modifications (e.g., consistency training) in order to understand their respective contributions relative to others.

**Limitations:**

yes

**Strengths And Weaknesses:**

The paper is overall clearly written with a fair degree of technical precision. Introducing a hybrid objective during training that includes both consistency and reconstruction losses seems like a very reasonable (albeit not novel) approach to decreasing inference time computational load. The authors introduce post training quantization for the residual streams in the compute heavy pairformer and DiT iterations where the linear part is maintained at high precision. The quantization effects seem well-explored (e.g., Figure 2).

Table 1 seems to include many "older" comparisons, absent AF3 or Boltz. Some comparison to SOTA would be helpful.

123: I don't think "mean flow" wasn't introduced by Lu & Song.

---

> ### Author Rebuttal · Authors · 2026-03-28
>
> We sincerely thank the reviewer for the positive assessment and constructive feedback. We appreciate the acknowledgment of our contributions. We address the three specific points below.
>
> ---
>
> **A1: Mean Flow citation.**
> Thank you for catching this. Mean Flow was introduced by Geng et al. (2025) [1], not Lu & Song. We will correct this citation in our final version.
>
> ---
>
> **A2: Table 1 missing comparisons with AF3/Boltz/Chai‑1, etc.**
> We acknowledge that direct comparison with recent co‑folding models would be valuable. However, such comparisons require careful contextualization due to differences in training data. These models (e.g., AlphaFold3, Boltz‑1/2, Protenix, SeedFold, Chai‑1) are trained on the full RCSB PDB (millions of protein-ligand interfaces), whereas our model is trained on the substantially smaller PDBBind2020 (~18k training complexes). Data scaling is a dominant factor in molecular docking—with sufficient data and strong pocket priors, both co‑folding models and our approach achieve 90–96% success rates on PoseBusters (e.g., PhysDock Figure 2, AlphaFold3[2] Extended Data Table 1).
>
> For reference, existing co‑folding models typically report 75–80% success rates on PoseBusters under their standard evaluation protocols. Our current model (trained on PDBBind2020, Linear kernel, NFE=20) already achieves **84.1%** on PoseBusters (as reported in response to Reviewer 1), demonstrating strong generalization despite the smaller training set.
>
> To enable a fairer comparison, we are currently training a scaled‑up version of our model on a larger dataset (including the full PDB) using PCR. This model will have a parameter count comparable to AlphaFold3 (~0.3B) while maintaining sub‑second inference speed, achieving competitive accuracy without sacrificing our efficiency advantages. We will release the results and the related agentic skills soon. Furthermore, we are pleased to make the model available to the open‑source community once the remote MCP service is launched. We believe that embedding real‑time deep learning‑based molecular docking tools into agentic workflows can bring transformative benefits to practical AI‑driven drug design.
>
> ---
>
> **A3: Ablations to understand individual contributions.**
> We fully agree and have provided such ablations. Below we summarize the inference cost decomposition (detailed in response to Reviewer 1, Q1), with all measurements on NVIDIA H20, 256‑token complex, 5 conformations:
>
> | Configuration | Pairformer (48 blk, NFE=1) | DiT (24 blk, NFE=1) |
> |---------------|---------------------------|---------------------|
> | TF32 | 514.44 ms | 165.91 ms |
> | BF16 | 396.72 ms | 127.25 ms |
> | BF16 + torch.compile | 229.44 ms | 63.67 ms |
> | W8A8 + torch.compile | 191.67 ms | 54.90 ms |
>
> **Key insights:**
> - **MSA removal** primarily simplifies feature input; the MSA track itself contributes marginally (Evoformer with MSA: 531.20 ms vs. Pairformer: 514.44 ms for 48 blocks). The computational bottleneck lies in triangular attention, triangular update, and PairBlock FFN (pair track accounting for over 85% of inference cost).
> - **PCR** reduces DiT NFE from 200 to 1–5, delivering the largest overall speedup.
> - **Quantization** provides modest latency gain but reduces memory by ~50%, enabling larger batch sizes—critical for high‑throughput screening.
>
> **Ablation of $\mathcal{L}_{\text{rec}}$:** Training without the reconstruction loss (pure consistency tuning) fails to converge stably, confirming its role as a critical trajectory anchor. This aligns with theoretical analysis in the consistency model literature [3][4][5].
>
> We will include these ablation details in the final manuscript.
>
> ---
>
> We are grateful for the thoughtful comments. We hope the clarifications and additional results strengthen the paper’s contribution and wish you all the best with your endeavors..
>
> ---
>
> ### References
> 1. Geng, Z., et al. (2025). Mean flows for one-step generative modeling. *NeurIPS*.
> 2. Abramson, J., et al. (2024). Accurate structure prediction of biomolecular interactions with AlphaFold 3. *Nature*.
> 3. Wang, F.-Y., et al. (2024). Stable Consistency Tuning. *arXiv:2410.18958*.
> 4. Song, Y., et al. (2024). Improved Techniques for Training Consistency Models. *ICLR*.
> 5. Issenhuth, T., et al. (2025). Improving Consistency Models with Generator-Augmented Flows. *ICML*.

---

> > ### Author Rebuttal · Reviewer_i1a5 · 2026-04-04
> >
> > Thank you for the response, I appreciate the ablations. I believe my assessment remains consistent with the current state of the paper.

---

> > > ### Author Response · Authors · 2026-04-06
> > >
> > > Thank you again for your thoughtful and constructive review. We sincerely appreciate your time, expertise, and valuable feedback.

---

### Official Review · Reviewer_DX7A · 2026-03-12

**Soundness:** 2
**Presentation:** 2
**Significance:** 2
**Originality:** 3
**Overall Recommendation:** 4
**Confidence:** 4

**Summary:**

This paper addresses the high inference latency of diffusion-based molecular docking models, proposing an efficient framework potentially applicable to real-time agentic workflows. The three core contributions are: (1) Progressive Consistency Regularization (PCR), which compresses diffusion sampling to 1–5 steps via curriculum scheduling while retaining a reconstruction loss to prevent quality erosion; (2) Residual Quantization, applying W8A8 quantization to linear projections in the Pairformer and DiT while keeping skip connections in BF16; (3) a theoretical unification of EDM and Flow Matching.

**Compliance With Llm Reviewing Policy:**

Affirmed.

**Final Justification:**

the rebuttal partially addresses my concerns.

**Key Questions For Authors:**

1. What prior information was provided to the baselines in Table 1 during evaluation?

2. What are the docking success rate and Chi-MAE when $\mathcal{L}_{\text{rec}}$ is removed?

3. Can the authors provide latency comparisons against MSA-free methods under identical hardware and conformation sampling settings?

4. Can the authors separately report the latency contributions of quantization?

5. I suggest the authors discuss the relationship to DCFold [1], which also applies consistency models to accelerate AlphaFold3 but takes a different approach.

[1] ICLR 2026, CFold: Efficient Protein Structure Generation with Single Forward Pass

**Limitations:**

yes

**Strengths And Weaknesses:**

**Strength**

1. PCR is well-motivated. The incompatibility between JVP-based consistency constraints and Pairformer's activation checkpointing is a genuine technical obstacle. Adopting the Secant regime with curriculum scheduling to circumvent this is logically sound and potentially transferable to other heavy biological model architectures.
2. The reconstruction loss anchor is principled. Retaining $\mathcal{L}_{\text{rec}}$ during consistency tuning to prevent high-frequency detail loss is intuitively well-justified.
3. The quantization analysis is a useful reference for the structural biology modeling community.

**Weaknesses**

1. Fairness of the main comparison is unclear due to potential prior leakage. The reported 83.18% uses Level 4 priors, while Appendix E.2 shows a ~10% absolute gain from Level 3 to Level 4. Since the paper does not specify what prior level the baselines in Table 1 use, it is hard to determine whether the comparison is fair.
2. Latency baselines are systematically biased. All baselines in Figure 3 (AlphaFold3, Chai-1, Boltz-2, etc.) are MSA-dependent, whereas this work is MSA-free. Comparisons against MSA-free methods (e.g., Protenix-Mini, ESMFold + docking module) are absent. The >300× speedup claim likely reflects MSA retrieval and processing costs in AF3 rather than architectural efficiency alone.
3. The contribution of $\mathcal{L}_{\text{rec}}$ is not quantified. PCR closely resembles ECT in its core design. The primary difference is the additional $\mathcal{L}_{\text{rec}}$ term, yet no ablation is provided to isolate its contribution.
4. The latency contribution of quantization is unreported. Figure 3 presents only the combined latency of PCR + quantization. The independent speedup attributable to quantization alone is never isolated.
5. The key theoretical claim is insufficiently supported. The claim that "performance upper bound is determined by feature richness rather than model complexity" lacks a critical ablation: systematically scaling model size under fixed features. The leap from "specific architectural nuances did not help" to "model complexity is generally unimportant" is not logically warranted.
6. Minor issues with framing. The MCP/Vibe Researching motivation is not backed by any implementation or agent-scenario evaluation, remaining largely rhetorical.

---

> ### Author Rebuttal · Authors · 2026-03-28
>
> We sincerely thank the reviewer for the detailed critique. We appreciate the acknowledgment of our technical contributions and address each point below.
>
> ---
>
> **A1: Prior information of baselines and fairness of comparison.**
> We thank the reviewer for raising this critical issue. Practically, nearly all deep learning docking methods incorporate some form of prior information, whether implicitly through training data bias, explicit pocket annotations, or architectural inductive biases. For example, blind docking methods (e.g., DiffDock) assume no pocket location but still rely on receptor geometric priors (e.g., full‑atom coordinates); co‑folding models rely on coevolutionary signals from MSA to infer pocket information; semi‑flexible methods like SurfDock are provided with detailed surface meshes, ligand center, and pocket residues—effectively Level 4 prior. The literature lacks a standardized framework to disclose these differences, making fair comparison notoriously difficult.
>
> To address this, we introduced the information leakage hierarchy (Appendix E.2, Table 3). Under the same Level 4 prior, our method (Linear kernel, NFE=20) achieves **84.1%** on PoseBusters, surpassing SurfDock at 81.5%. Under the same Level 1 prior (blind docking), our method achieves **49.1%** on PoseBusters, surpassing DiffDock at 42.1%. Our hierarchy thus serves both as a diagnostic tool and a community call for standardized reporting. We will explicitly clarify the prior settings of each baseline in the final version.
>
> Importantly, our model uses only Cα distograms (similar to AlphaFold2 template) as geometric priors. This is not a strict semi‑flexible docking approach, but a deliberate efficiency–accuracy trade‑off. We acknowledge this limitation and discuss ensemble docking as an effective mitigation.
>
> ---
>
> **A2: Docking success rate and Chi‑MAE when $L_{\text{rec}}$ is removed.**
> Without $L_{\text{rec}}$ (pure consistency tuning), the model fails to converge stably. This aligns with theory: Wang et al. [1] show consistency training (CT) uses high‑variance single‑sample estimation, causing instability, while distillation (CD) benefits from a stable teacher. Song et al. [2] note CD is limited by teacher quality, but CT is harder to optimize. Issenhuth et al. [3] prove CT’s estimation error persists in continuous time. $L_{\text{rec}}$ acts as a trajectory anchor, stabilizing training.
>
> ---
>
> **A3: Latency comparisons against MSA‑free methods & benchmark evaluation.**
> All Figure 3 numbers are GPU inference times only, excluding MSA retrieval. The bottleneck lies in triangular attention, triangular update, and PairBlock SwiGLU FFN (>85% cost). The MSA track itself contributes marginally:
>
> | Module | Latency (48 blk, NFE=1) |
> |--------|------------------------|
> | Evoformer (with MSA) | 531.20 ms |
> | Pairformer (MSA‑free) | 514.44 ms |
>
> Removing MSA primarily simplifies feature input. To address the benchmark concern, we have evaluated on PoseBusters [4]. Our model achieves **84.1%** , outperforming SurfDock at **81.5%** .
>
> ---
>
> **A4: Separate latency contribution of quantization.**
> Quantization contribution on H20:
>
> | Configuration | Pairformer (48 blk, NFE=1) | DiT (24 blk, NFE=1) |
> |---------------|---------------------------|---------------------|
> | BF16 + torch.compile | 229.44 ms | 63.67 ms |
> | W8A8 + torch.compile | 191.67 ms | 54.90 ms |
>
> Primary benefit is **memory reduction** (~50%), enabling larger batch sizes for high‑throughput screening.
>
> ---
>
> **A5: Relationship to DCFold.**
> DCFold distills an existing heavy model (Protenix); our PCR is a **from‑scratch** strategy integrating consistency regularization into diffusion pre‑training. Key differences:
>
> | Aspect | DCFold | Ours (PCR) |
> |--------|--------|------------|
> | Base model | Distills pre‑trained Protenix | Trained from scratch |
> | Consistency approach | Distillation with reduced recycling | PCR with curriculum learning |
> | Reconstruction loss | Not emphasized | **Key component**: $\mathcal{L}_{\text{rec}}$ |
> | Need teacher model | Yes | No |
>
> ---
>
> **On MCP/Vibe Researching framing.**
> We will tone this down and focus on concrete contributions. We are packaging our model as a lightweight skill that can be invoked via natural language (e.g., “dock this SMILES into pocket of PDB 1XYZ”) with real-time response. A remote MCP server will be released shortly.
>
> ---
>
> We are grateful for the rigorous critique. We believe these clarifications address the concerns and substantially strengthen the paper. We hope this response merits reconsideration for acceptance. We wish you all the best with your endeavors.
>
> ---
>
> ### References
> 1. Wang, F.-Y., et al. (2024). Stable Consistency Tuning. *arXiv:2410.18958*.
> 2. Song, Y., et al. (2024). Improved Techniques for Training Consistency Models. *ICLR*.
> 3. Issenhuth, T., et al. (2025). Improving Consistency Models with Generator-Augmented Flows. *ICML*.
> 4. Buttenschoen, M., et al. (2024). PoseBusters. *Chem. Sci.*

---

### Official Review · Reviewer_gyn6 · 2026-03-13

**Soundness:** 4
**Presentation:** 3
**Significance:** 3
**Originality:** 2
**Overall Recommendation:** 5
**Confidence:** 3

**Summary:**

This work aims to make diffusion-based molecular docking models fast enough for real-time use in agentic scientific workflows (e.g., via MCP). The authors propose three contributions: a streamlined co-folding architecture (Pairformer and Dual-Scale DiT) that strips away components they find unnecessary; a curriculum-based consistency distillation method that drastically reduces the required number of function evaluations called Progressive Consistency Regulation (PCR); Residual Quantization, a mixed precision scheme applied post-training. The authors demonstrate a >300x speedup over AlphaFold3 on an H20 GPU while achieving SOTA docking success rates.

**Compliance With Llm Reviewing Policy:**

Affirmed.

**Final Justification:**

Authors addressed all concerns in their rebuttal and in their latest response shared their ablasion study w.r.t. their curriculum approach. Through their clarifications, they increased the soundness of their paper, and persuaded me therefore recommend the paper for acceptance.

**Key Questions For Authors:**

- how much of the speedup comes from fewer NFE via PCR, quantization, JIT/torch.compile, and architectural simplification (removing MSA track, etc.) individually?
- Can you provide cross-docking or blind-docking results?
- Why was the approach not evaluated on other benchmarks?
- How does your method compare to other co-folding competitors?

**Limitations:**

Yes

**Strengths And Weaknesses:**

The paper and experimental results are technically sound and comprehensive. Ablation studies on kernel choice, NFE count, and quantization precision seem well-designed and are informative. The main critique point would be that the evaluation is limited to a single benchmark (PDBBind2020 re-docking).

The paper is well written and has a clear narrative. The framing around "vibe researching" is timely, but has a bit of a marketing vibe. The paper tries to sell the research a bit too much: "enabling large-scale industrial virtual screening with unprecedented throughput," "we present a vertical foundation model engineered to dismantle the computational bottleneck," "theoretically elegant but empirically superior," "achieves a 'sweet spot', delivering robust SOTA accuracy with sub-second latency." A little less hype and a few extra benchmarks would persuade me more of their methods superiority and robustness. The paper is doing a lot, and it appears at times that it was hard to fit everything that had to be present into the main-body of the paper. For example the curriculum learning schedule choice is introduced, but not justified.

The paper doesn't clarify which gains come from which component, since the speedup the combined effect of all methods introduced, but it's not possible to gauge the impact of each individual method: architecture simplification, PCR, quantization, JIT compilation.

The problem tackled by the paper is significant and important. The speedup is impressive and practically relevant. However, it's questionable how this work translates to real world applications due to the limited re-docking only experiments and the semi-flexible docking assumption (doesn't capture large-scale shape changes). The paper would be more impactful if it demonstrated its performance on a more challenging or realistic task.

The originality of this paper consists mainly in the combination of multiple techniques and the PCR formulation with curriculum learning.

Typo: should "FinalLayer" in section header E.3 be "Final Layer"?

---

> ### Author Rebuttal · Authors · 2026-03-28
>
> We sincerely thank the reviewer for recognizing the technical soundness and practical relevance of our work. Below we address each point.
>
> ---
>
> **A1: Speedup decomposition.**
> All measurements are GPU inference time on a single NVIDIA H20 with a 256‑token complex generating 5 conformations (max token length is 256, corresponding to up to 2048 atoms; shorter sequences are padded). Our reported 0.17 s latency is GPU time only, excluding CPU preprocessing.
>
> In a standard AlphaFold3‑like architecture, the Pairformer (10 NFE) and Token‑wise DiT (200 NFE) dominate inference cost. Prior works (PhysDock, DCFold) show that setting Pairformer NFE to 1 suffices for docking. Below we report GPU cost of standard Pairformer and Token‑wise DiT under different configurations:
>
> | Configuration | Pairformer (48 blk, NFE=1) | Token‑wise DiT (24 blk, NFE=1) |
> |---------------|---------------------------|-------------------------------|
> | TF32 | 514.44 ms | 165.91 ms |
> | BF16 | 396.72 ms | 127.25 ms |
> | BF16 + torch.compile | 229.44 ms | 63.67 ms |
> | W8A8 + torch.compile | 191.67 ms | 54.90 ms |
>
> **Note:** The computational bottleneck of Pairformer lies in pair track (accounting for over 85% of inference cost).
>
> The **combination** of model simplification (Pairformer reduced from 48 to 8 blocks, Token‑wise DiT reduced from 24 to 6 blocks, Pairformer NFE set to 1, and Token‑wise DiT NFE reduced from 200 to 1–5 via PCR), low‑precision inference (BF16, W8A8), and torch.compile enables full‑atom generation in 0.17 s. It should be noted that this time also includes the overhead of other components such as the embedding layer, final layers, etc.
>
> ---
>
> **A2: Cross‑docking and blind docking results.**
> **Cross‑docking.** Following the PhysDock protocol on 25 proteins, our model achieves **56% (14/25)** , outperforming SurfDock at **44% (11/25)** , confirming generalization across receptor conformations.
>
> **Blind docking (no pocket prior).** On PDBBind2020 test dataset (Level 1 prior), our model (Linear kernel, NFE=20) achieves **46.7%** (RMSD < 2.0 Å), comparable to DiffDock (47.7%). On PoseBusters, our blind docking performance is **49.1%** , surpassing DiffDock at **42.1%** .
>
> The lower accuracy of blind docking underscores one of  our key message: **pocket prior strength heavily influences docking success**, which we explicitly highlight in Appendix Table 3.
>
> ---
>
> **A3: Evaluation on additional benchmarks.**
> We primarily used PDBBind2020 test dataset for direct comparability with existing baselines. We have now evaluated on **PoseBusters** [1], a gold‑standard re‑docking benchmark. Our model achieves **84.1%** , outperforming SurfDock at **81.5%** . These results confirm strong generalization and will be included in the final version.
>
> ---
>
> **A4: Comparison with co‑folding competitors.**
> We highlight three key distinctions:
>
> **1. Coevolution‑free vs. coevolution‑dependent.** Many co‑folding models (AlphaFold3, Boltz‑2, SeedFold, Protenix) rely on deep MSAs or PLMs (protein language models). While coevolution informs protein structure, its relevance to docking is limited—binding geometry is governed by local atomic interactions, not evolutionary signals. Removing MSA and PLM features forces our model to learn from structural and geometric priors alone, enabling a more physically grounded approach.
>
> **2. Feature richness > architecture complexity.** Appendix E.2 shows that with sufficiently rich interface priors (e.g., approximate pocket center and residue‑level contact flags), even a simplified architecture achieves SOTA performance. We therefore adopt full‑atom attention to capture atomic‑level interactions, focusing compute on atom spatial geometry.
>
> **3. Decoupling interface localization from pose refinement.** Following insights from IsoDDE [2], we treat docking as a two‑stage process: (a) identify binding site, (b) refine atomic pose. This modular design enables natural language alignment, lightweight deployment, and independent optimization of each stage.
>
> In contrast to heavy co‑folding competitors that solve interface identification and atomic refinement jointly, our approach focuses on pose refinement under reasonable physical priors—yielding the efficiency required for low‑latency applications while maintaining competitive accuracy.
>
> ---
>
> **On writing style and typo.** We agree to tone down promotional language and will revise the introduction and conclusion accordingly. We will correct “FinalLayer” to “Final Layer.”
>
> ---
> We are grateful for the detailed feedback and believe these clarifications strengthen the paper. We hope this response addresses the concerns and meets the expectations for acceptance.  We wish you all the best with your endeavors.
>
> ---
>
> ### References
> 1. Buttenschoen, M., et al. (2024). PoseBusters. *Chem. Sci.*
> 2. [IsoDDE Technical Report](https://storage.googleapis.com/isomorphiclabs-website-public-artifacts/isodde_technical_report.pdf)

---

> > ### Author Rebuttal · Reviewer_gyn6 · 2026-04-04
> >
> > Thank you for your response. The authors addressed most of my concerns. The authors explain why a comparison to other cofolding methods is difficult but don't provide such a comparison.
> >
> > Follow up:
> > I still would appreciate a clarification as to why the specific design choice for the curriculum schedule was made. Were other schedules tested? Was there an ablation study?

---

> > > ### Author Response · Authors · 2026-04-05
> > >
> > > We thank the reviewer for the thoughtful follow-up. To systematically justify our curriculum schedule design, we have conducted two ablation studies: (1) convergence analysis of different consistency training configs, and (2) the effect of maximum discretization steps $N_{max}$. Below we present the results and analysis.
> > >
> > > ---
> > >
> > > **Ablation 1: Convergence Analysis of Training Configurations**
> > >
> > > | Case | Loss | $\Delta t$ | Curriculum? | Other Config | Model | Convergence |
> > > |-|-|-|-|-|-|-|
> > > | 0 | $L_{cons}$ | N/A | N | JVP, Continuous CM | sCM, MF | - |
> > > | 1 | $L_{cons}$ | Fixed | N | - | CM(CT) | Not converge |
> > > | 2 | $L_{cons}$ | Fixed | N | Positional Time Embeddings | Early sCM | Not converge |
> > > | 3 | $L_{cons}$ | Progressive | Y | - | iCT base | Not converge |
> > > |4| $L_{cons}$ | Fixed | N | Need Pretraining | ECT 1-Hour Prototyping | Initial converge then diverge |
> > > |5| $L_{cons}+L_{rec}$ | Fixed | N | - | SCT/ECT base | Extremely slow |
> > > |6| $L_{cons}+L_{rec}$ | Progressive | Y | EMA decay on $L_{rec}$ |- | Initial converge then diverge |
> > > |Ours | $L_{cons}+L_{rec}$ | Progressive| Y | - | - |**Converges**|
> > >
> > > $L_{cons}$: consistency loss; $L_{rec}$: reconstruction loss; Y: Yes; N: No
> > >
> > > **Detailed analysis for each case:**
> > >
> > > - **Case 0 (Continuous CM):** Aims to model tangents for continuous-time consistency. However, JVP computation is not natively optimized in our framework, causing prohibitive GPU memory (OOM).
> > > - **Case 1 (CM(CT)):** The model bootstraps on a constantly changing target without a stable reference, failing to establish noise-to-data mapping [SCT].
> > > - **Case 2 (Early sCM):** Positional embeddings are smoother than Fourier alternatives in sCM. Yet, without $L_{rec}$ anchoring, the model still cannot converge.
> > > - **Case 3 (iCT base):** Progressive $\Delta t$ without reconstruction regularization triggers the **“Curse of Consistency”** [ECT], leading to divergence.
> > > - **Case 4 (ECT 1-Hour Prototyping):** Fixed minimal $\Delta t$ provides no curriculum. On complex point clouds, gradient flows become highly imbalanced — supervision at large noise levels is extremely weak. The model shows initial progress but diverges due to numerical instability [ECT].
> > > - **Case 5 (SCT/ECT base):** Adding $L_{rec}$ prevents collapse, but fixed $\Delta t$ offers no coarse-to-fine learning. The model remains trapped in high-bias small-step region, unable to capture global trajectory geometry → extremely slow convergence [SCT].
> > > - **Case 6 (Experimental, progressive $\Delta t$ + EMA decay on $L_{rec}$):** Tests whether the model can shed the reconstruction anchor. As $L_{rec}$ decays, the physical coordinate anchor weakens, causing gradient explosion.
> > > - **Ours (fixed $L_{rec}$ + progressive $\Delta t$ shrinkage):** **Self-anchoring effect**: $L_{rec}$ enforces boundary condition $f(x,0)=x$, providing a stable geometric reference. Progressive shrinkage enables coarse-to-fine learning: large Δt for global exploration, small Δt for local refinement.
> > >
> > > The importance of a fixed reconstruction anchor has been further demonstrated in recent works (e.g., SoFlow). It is worth noting that the one-step generated examples in the original CM and MF works exhibit noticeable physical implausibility, which can be attributed to the absence of fixed reconstruction anchor.
> > >
> > > ---
> > >
> > > **Ablation 2:** Effect of maximum discretization steps ($N_{max}$) on 1-step sampling success rate
> > >
> > > | $N_{max}$ | Success Rate (%) |
> > > |-|-|
> > > | 40 | 73.1 |
> > > | 100 | 75.9 |
> > > | 250 | 77.8 |
> > > | 500 | 79.2 |
> > > |1000 (ours)| **79.4** |
> > > |2000|76.2|
> > >
> > > (Curriculum schedule uses Karras noise scheduling. $N_{max}$ controls the finest discretization interval (smallest $\Delta t$). Model: DiT, NFE=1.)
> > >
> > > **Analysis:**
> > >
> > > - **Performance improvement (40 → 1000):** Increasing $N$ reduces discretization bias. Finer time intervals make secants approximate tangents more accurately, providing precise supervision to follow the PF-ODE trajectory [iCT, ECT].
> > >
> > > - **Performance degradation at $N_{max}=2000$:** Success rate drops from 79.4% to 76.2%. This is the **“Curse of Consistency”** [ECT]: when $N$ becomes too large ($\Delta t \to 0$), per-step prediction errors accumulate. If the decay of $e_{max}$ is slower than the growth of $N$, total error $\|f_\theta(x_T)-x_0\|$ increases. Additionally, precision limits cause floating-point errors to amplify over $N=2000$ steps, degrading physical plausibility.
> > >
> > > **Optimal range:** $N_{max}=500\text{–}1000$ balances bias reduction and error accumulation.
> > >
> > > ---
> > >
> > > Thank you again for the insightful suggestion. We hope these additional ablations and clarifications sufficiently address your concerns and would greatly appreciate your reconsideration of the score.
> > >
> > > ---
> > >
> > > **References List:**
> > > [CM] https://arxiv.org/abs/2303.01469
> > > [iCT] https://arxiv.org/abs/2310.14189
> > > [sCM] https://arxiv.org/html/2410.11081v1
> > > [MF] https://arxiv.org/abs/2505.13447
> > > [ECT] https://arxiv.org/abs/2406.14548
> > > [SCT] https://arxiv.org/abs/2410.18958
> > > [SoFlow] https://arxiv.org/abs/2512.15657

---

### Decision · Program_Chairs · 2026-04-30

**Decision:**

Accept (spotlight)

**Comment:**

This paper presents a quantized consistency diffusion framework for molecular docking designed to achieve sub-second inference latency, specifically tailored for real-time agentic workflows. By synergizing a streamlined Pairformer and Dual-Scale DiT architecture with Progressive Consistency Regularization (PCR) and mixed-precision Residual Quantization, the model achieves a remarkable speedup of over 300x compared to AlphaFold3 while maintaining state-of-the-art accuracy on the PDBBind 2020 dataset. The framework's strengths lie in its impressive computational efficiency, principled use of reconstruction loss as a trajectory anchor, and the elimination of redundant MSA-dependent components. While initial reviews noted weaknesses regarding the limited benchmark scope, the potential for prior leakage, and the absence of empirical agent-integration evidence, these were considered addressable during the discussion phase.

Therefore, the final recommendation is to accept the paper.